# Structural optimization of the excavator boom under extreme working conditions using EDEM–ADAMS coupled simulation

Huawei Wu[1,2,3], Xiaolong Ding[1,2]*, Gui Liu[1,2,3], Xiaoyuan Zhu[4], Hualiang Wang[5]

**1** Hubei Longzhong Laboratory, Hubei University of Arts and Science, Xiangyang, China, **2** Hubei Key Laboratory of Pure Electric Vehicle Power System Design and Testing, Hubei University of Arts and Science, Xiangyang, China, **3** Chinese Mechanical Engineering Society, Beijing, China, **4** College of Mechanical Engineering, Southeast University, Nanjing, Jiangsu, China, **5** Xiangyang Zhongliang Construction Machinery Co., Ltd., Xiangyang, China

* dingxiaolong601@gmail.com

## Abstract

Aiming at the extreme loading conditions of an backhoe loader under dynamic and uncertain environments, this paper proposes a structural optimization method based on EDEM-ADAMS for a compact multi-functional excavating transporter device. The D-H coordinate kinematic analysis model of the backhoe loader is constructed, and the forward kinematic solution is obtained. Using the multi-body dynamics software ADAMS and enhanced discrete element software, a virtual prototype model and a discrete element material model are established. Through the coupled simulation method of EDEM-ADAMS, the load distribution of two working modes, forward digging and side digging, is analyzed, and the extreme working conditions of the boom are determined. Finally, topology optimization of the boom under extreme working conditions is performed to strengthen the local structure. The results show that after optimization, the boom's mass is reduced by 14.32 kg (13.80%), the maximum stress is reduced by 26.12%, and the total deformation is reduced by 29.11%. Compared to existing optimization methods, the equivalent stress and total deformation of the proposed optimized model are reduced by 18.76% and 22.27%, respectively. These improvements not only achieve weight reduction but also significantly enhance the structural strength and safety. The optimized design has significant implications for the structural optimization of similar backhoe loader under extreme working conditions.

## I. Introduction

Hydraulic excavators serve as critical engineering machinery extensively deployed in mining, construction, agriculture, and infrastructure sectors [1]. Their excavation mechanisms operate in complex environments prone to failures, including

**Data availability statement:** All relevant data for this study are publicly available from the ScienceDB repository (https://doi.org/10.57760/sciencedb.26309).

**Funding:** National Natural Science Foundation of China under Grant;Award Number:52472405;Grant Recipient:Huawei Wu,Ph.D.(2):Find a Funder:Natural Science Foundation of Hubei Province;Award Number:2024AFB219;Grant Recipient:Huawei Wu,Ph.D.(3)Find a Funder:Special Project of Central Government for Local Science and Technology Development of Hubei Province;Award Number:2024CSA081.

**Competing interests:** The authors have declared that no competing interests exist.

deformation and fatigue fractures induced by alternating stresses in booms and bending stresses in sticks, wear-induced failures at connecting-pin articulations, and bucket damage from impact and abrasion [2,3].

To address these challenges, significant research has focused on excavation mechanism design. Feng Hao [4] et al. developed a dynamic excavator model, employing the Newmark algorithm in MATLAB for dynamic load computation with experimental validation. Yu [5] et al. established a parametric full-machine finite element model, simulating hydraulic cylinder behavior via multipoint constraint and linkage elements while proposing a numerical method for pin-joint analysis to optimize boom structures. Cao Yuanwen [6] et al. implemented ADAMS-based virtual prototyping for multifunctional quick couplers, enhancing design efficiency through simulation. Sun Haoran [7] et al. introduced a bucket lightweight design methodology correlating continuous and ultimate digging forces, developing APDL-based parametric models. Suryo H.S [8]. and Hadi S.S [9]. applied topology optimization and FEM to CAT 374D L-boom excavators and bucket teeth respectively, achieving mass reduction within allowable stress constraints. Zhao Zhengyu [10] et al. conducted Altair Inspire-driven topology optimization for excavator secondary arms under stiffness-strength constraints, validating redesigned components through 3D printing.

While these studies characterize kinematic behaviors through theoretical and simulation approaches, they oversimplify operational scenarios by neglecting granular material interactions, limiting simulation fidelity.

Recent advances in virtual prototyping facilitate coupled dynamics-discrete element modeling for excavator performance analysis. Bi Qiusi [11] et al. integrated ADAMS-EDEM to quantify digging resistance in shovel operations. Wang Tongjian [12] et al. reconstructed excavation states using RecurDyn-EDEM with field-data calibration. Wu Di [13] et al. validated ADAMS-EDEM excavation resistance predictions against strain-gauge measurements while analyzing soil parameter effects. Liu Chang [14] et al. extracted spatiotemporal bucket force distributions via DEM-MBD coupling.

DEM simulations demonstrate exceptional consistency with physical load measurements, validating model reliability while reducing experimental costs. Nevertheless, current research predominantly focuses on excavation resistance analysis, lacking systematic methodologies for: multi-condition load spectrum generation, critical service condition identification, and ultimate strength assessment.

This study bridges these gaps by establishing a kinematic-structural response framework to:

i.  Develop multi-scenario load modeling

ii.  Precisely identify critical working conditions

iii.  Quantify extreme load distributions

iv. Propose concurrent strength-stiffness optimization for structural enhancement.

## II. **Kinematic analysis and dynamic modeling of excavation mechanism**

**A.  Prototype development.**  Traditional mechanical equipment has limited functionality, and narrow construction sites frequently cause multi-machine

congestion that reduces efficiency. To address these issues, this paper designs a multifunctional small excavation-transportation machine and proposes an optimization method for extreme working conditions. This equipment (see Fig 1) can perform tasks including material shoveling, short-distance transportation, trench cleaning, and rescue operations, adapting to variable construction requirements. Its front excavation mechanism enables lateral operation through left and right slewing supports, while materials are transported by a rear scraper and conveyor belt system.

**B. Kinematic analysis of excavation mechanism.** The mechanical excavation mechanism comprises a boom, stick, and bucket, articulated to form a three-linkage mechanism operating exclusively in the vertical plane. The kinematic model is constructed using the D-H method, with D-H parameters defining coordinate systems and variables between linkages.

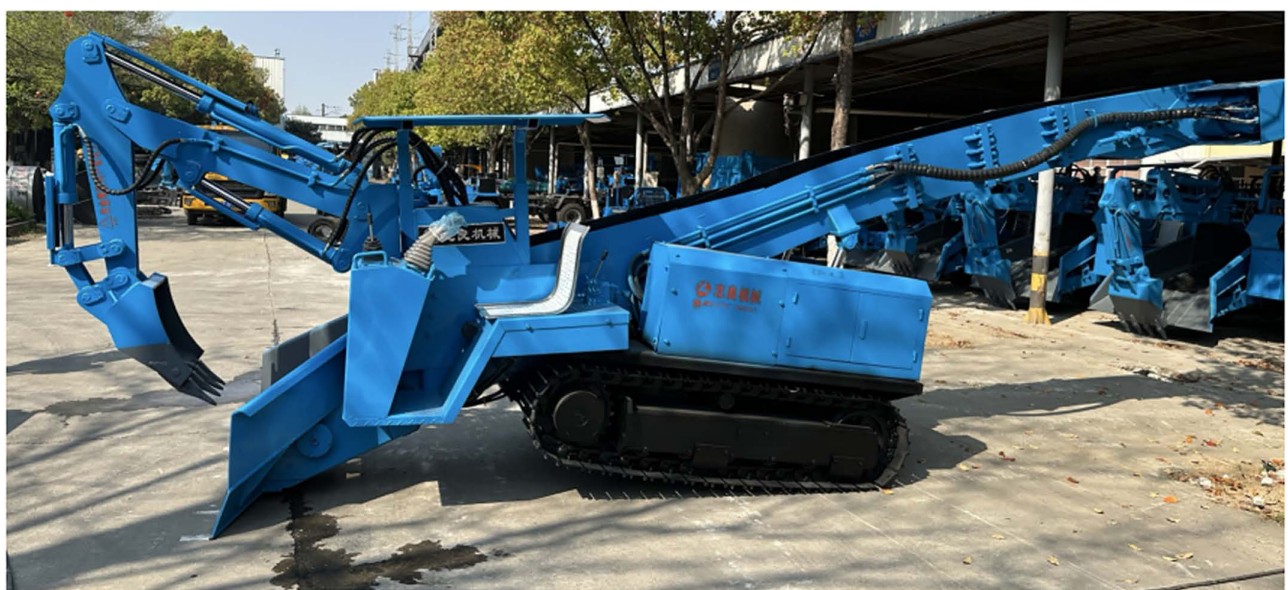

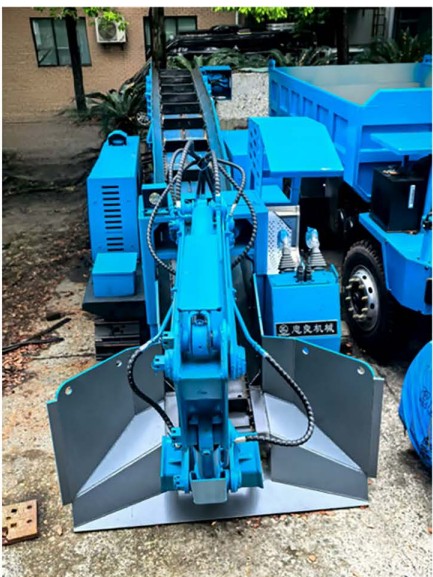

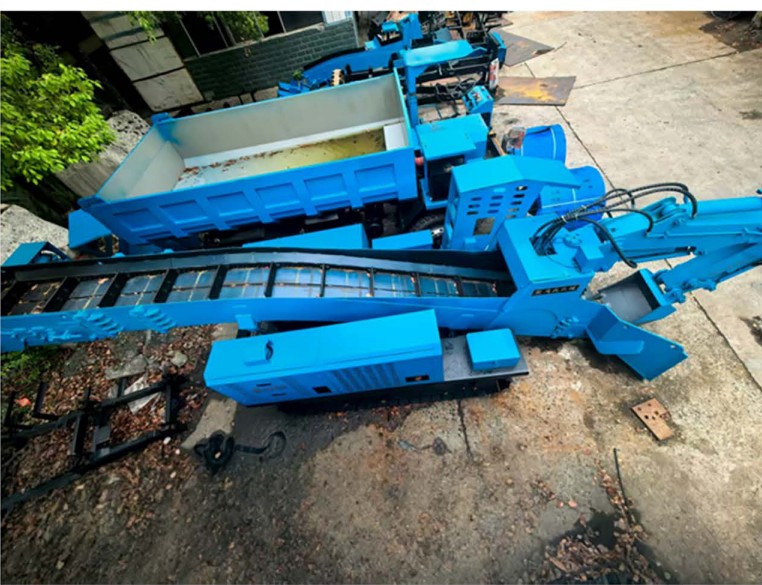

**Fig 1. Structural assembly of the backhoe loader.** (a) Side view. (b) Front view. (c) Top view.

Structural parameters (*ai, ai, di*) specify fixed inter-joint distances, while kinematic parameter $\theta_i$ describes joint motion states. With the boom rotation center at point *O*, linkage coordinate systems {*O*},{*A*},and {*B*} are established at articulation points *O*, *A*, and *B* respectively along the hinge axis. System {*C*} is defined at the bucket's tail end *C*, as detailed in Fig 2.

The general transformation equation for the D-H coordinate linkage transformation matrix [15] is expressed as:

$$
{}^{i-1}_{i}T = \begin{bmatrix}
\cos\theta_i & -\cos\alpha_i\sin\theta_i & \sin\alpha_i\sin\theta_i & a_i\cos\theta_i \\
\sin\theta_i & \cos\alpha_i\cos\theta_i & -\sin\alpha_i\cos\theta_i & a_i\cos\theta_i \\
0 & \sin a_i & \cos\alpha_i & d_i \\
0 & 0 & 0 & 1
\end{bmatrix}
$$

(1)

To construct the D-H homogeneous transformation matrix, the D-H parameters for each linkage must first be determined. The key parameters characterizing each linkage are the linkage length (*ai*), defined as the distance along the common normal between adjacent joint axes, and the linkage twist (*ai*), defined as the angle about the common normal between adjacent joint axes. The relative displacement between two adjacent linkages is defined by the linkage offset (*di*), the distance along the joint axis, and the joint angle (*θi*), the angle about the joint axis. Analysis of these four parameters (*ai, ai, di, θi*) allows the D-H parameter table to be established in Table 1, forming the foundation for constructing the kinematic model.

Based on the established D-H parameter table, and considering that the pose variation of the backhoe loader is confined to the longitudinal plane, it follows that both $\alpha_{i-1}$ and $d_i$ become zero. Accordingly, the D-H linkage matrix of the robotic arm can be simplified to obtain the following simplified expression:

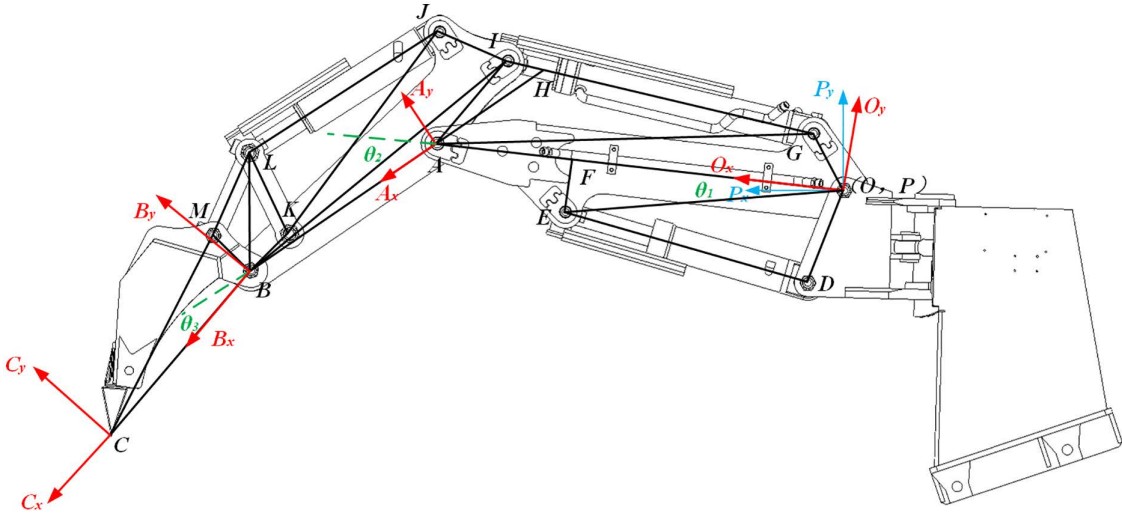

**Fig 2. Linkage coordinates of backhoe loader.**

**Table 1. D-H linkage parameter table.**

| i | $a_{i-1}$ | $\alpha_{i-1}$ | $d_i$ | $\theta_i$ |
|---|-----------|-----------------|-------|------------|
| 1 | OA | 0 | 0 | θ1(∠OxOA) |
| 2 | AB | 0 | 0 | θ2(∠AxAB) |
| 3 | BC | 0 | 0 | θ3(∠BxBC) |

$$\substack{P\\O}T = \begin{bmatrix} \cos\theta_1 & -\sin\theta_1 & 0 \\ \sin\theta_1 & \cos\theta_1 & 0 \\ 0 & 0 & 1 \end{bmatrix}$$

(2)

$$\substack{O\\A}T = \begin{bmatrix} \cos\theta_2 & -\sin\theta_2 & OA \\ \sin\theta_2 & \cos\theta_2 & 0 \\ 0 & 0 & 1 \end{bmatrix}$$

(3)

$$\substack{A\\B}T = \begin{bmatrix} \cos\theta_3 & -\sin\theta_3 & AB \\ \sin\theta_3 & \cos\theta_3 & 0 \\ 0 & 0 & 1 \end{bmatrix}$$

(4)

$$\substack{B\\C}T = \begin{bmatrix} 1 & 0 & BC \\ 0 & 0 & 0 \\ 0 & 0 & 1 \end{bmatrix}$$

(5)

Following the D-H method, the homogeneous transformation matrix from the base frame {P} to the bucket-end frame {C} is derived through the product of consecutive link transformation matrices:

$$\substack{P\\C}T = \begin{bmatrix} \cos(\theta_1+\theta_2+\theta_3) & -\sin(\theta_1+\theta_2+\theta_3) & OA\cos\theta_1 + AB\cos(\theta_1+\theta_2) + BC\cos(\theta_1+\theta_2+\theta_3) \\ \sin(\theta_1+\theta_2+\theta_3) & \cos(\theta_1+\theta_2+\theta_3) & OA\sin\theta_1 + AB\sin(\theta_1+\theta_2) + BC\sin(\theta_1+\theta_2+\theta_3) \\ 0 & 0 & 1 \end{bmatrix}$$

(6)

By analyzing the operational posture of the cutting and backhoe loader, we calculate the hinge point coordinates and rotation angles $\theta_1$, $\theta_2$, $\theta_3$. Combined with the linkage parameters, the absolute coordinates of all nodes on the robotic arm are then precisely determined.

**C. Forward kinematics formulation.** The forward kinematics solution derives the pose parameters (position and orientation) from kinematic inputs, typically joint variables. For the backhoe loader this is achieved by establishing constraint equations governing the geometric relationships between adjacent links. Specifically, the stroke lengths of three hydraulic cylinders—corresponding to actuating links $L_1, L_2, L_3$—serve as inputs to solve the simultaneous equations for joint angles $\theta_1$, $\theta_2$, and $\theta_3$.

i. **Forward kinematic computation for $\theta_1$**

Fig 3 illustrates the hydraulic circuit of the boom cylinder system, where $\theta_1$ denotes $\angle FOP_x$. Within this circuit, $OD$, $OE$, $OF$, and $EF$ represent fixed structural parameters, while $DE$ corresponds to the hydraulic cylinder input variable $L_1$ (stroke length). Applying the law of cosines to $\Delta DOE$ and $\Delta EOG$ yields, we derive:

$$\angle DOF = \angle DOE + \angle EOF = \arccos\frac{DO^2 + EO^2 - L_1{}^2}{2DO \times EO} + \arccos\frac{EO^2 + FO^2 - EF^2}{2EO \times FO}$$

(7)

This results in the fundamental relation:

$$\theta_1 = \angle FOP_x = \angle DOF - \angle DOP_x = \arccos\frac{DO^2 + EO^2 - L_1{}^2}{2DO \times EO} + \arccos\frac{EO^2 + FO^2 - EF^2}{2EO \times FO} - \angle DOP_x$$

(8)

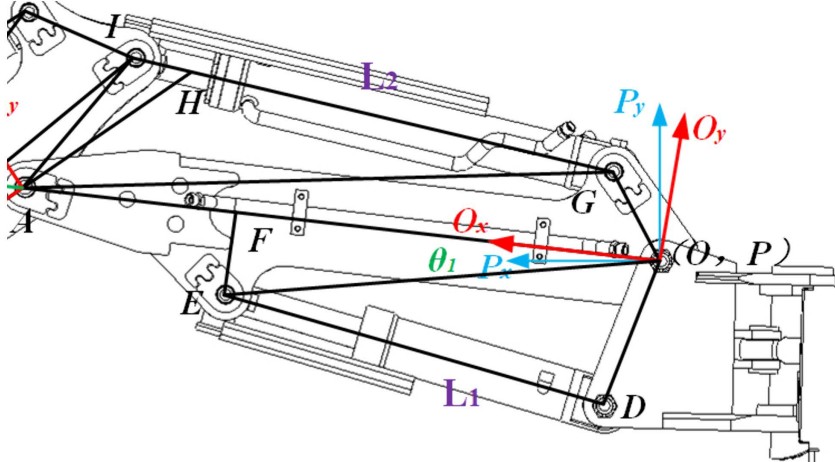

**Fig 3. D-H linkage diagram of boom.**

## ii. Forward kinematic computation for $\theta_2$

The bucket rod hydraulic cylinder circuit system is shown in Fig 4, and the required $\theta_2$ is ∠HAO. in the movable arm hydraulic circuit, *OG*, *OA*, *AG*, *IA*, *IB*, *AB* are known structural parameters, and *IG* is the hydraulic cylinder input $L_2$.

The collinearity of points *H*, *A*, and *B* implies that:

$$\angle IAH = \pi - \angle IAB \tag{9}$$

Therefore, applying the law of cosines to Δ*BAI* and Δ*AGI* yields, we derive:

$$\angle HAG = \angle IAG - \angle IAH = \arccos \frac{IA^2 + GA^2 - L_2^2}{2IA \times GA} - \pi + \arccos \frac{IA^2 + BA^2 - IB^2}{2IA \times BA} \tag{10}$$

Combining these governing relations yields and applying the law of cosines to Δ*AOG*, we derive:

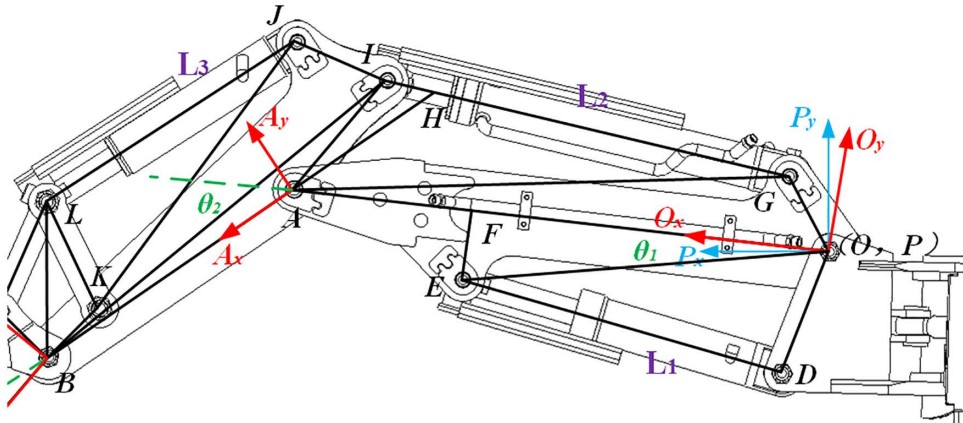

**Fig 4. Bucket arm D-H linkage diagram.**

$$\theta_2 = \angle\text{HAO} = \angle\text{HAG} + \angle\text{GAO} = \arccos\frac{\text{IA}^2 + \text{GA}^2 - L_2{}^2}{2\text{IA} \times \text{GA}}$$
$$+arccos\frac{\text{IA}^2 + \text{BA}^2 - \text{IB}^2}{2\text{IA} \times \text{BA}} + \arccos\frac{\text{GA}^2 + \text{OA}^2 - \text{GO}^2}{2\text{GA} \times \text{OA}} - \pi \tag{11}$$

### iii. Forward kinematic computation for $\theta_3$

Fig 5 depicts the hydraulic circuit of the bucket cylinder system, where $\theta_2$ is defined as $\angle CBN$. Within the bucket hydraulic circuit, $JK$, $LK$, $JB$, $JA$, $BK$, $LM$, $BM$, $MC$, and $BC$ constitute fixed structural parameters, while $LJ$ represents the hydraulic cylinder input variable $L_3$ (stroke length).

Applying the law of cosines to $\Delta LBJ$ yields:

$$\angle\text{LBJ} = \angle\text{LBK} - \angle\text{BJK} = \arccos\frac{\text{JK}^2 + L_3{}^2 - \text{LK}^2}{2\text{JK} \times L_3} - \arccos\frac{\text{BJ}^2 + \text{JK}^2 - \text{BK}^2}{2\text{BJ} \times \text{JK}} \tag{12}$$

$$\text{LB} = \sqrt{\text{LJ}^2 + \text{BJ}^2 - \cos\angle\text{LJB} \times 2\text{LJ} \times \text{BJ}} \tag{13}$$

Applying the law of cosines to $\Delta JBA$ yields:

$$\angle\text{JBA} = \arccos\frac{\text{JB}^2 + \text{AB}^2 - \text{JA}^2}{2\text{JB} \times \text{AB}} \tag{14}$$

$$\angle\text{MBN} = \pi - \angle\text{MBL} - \angle\text{LBJ} - \angle\text{JBA} \tag{15}$$

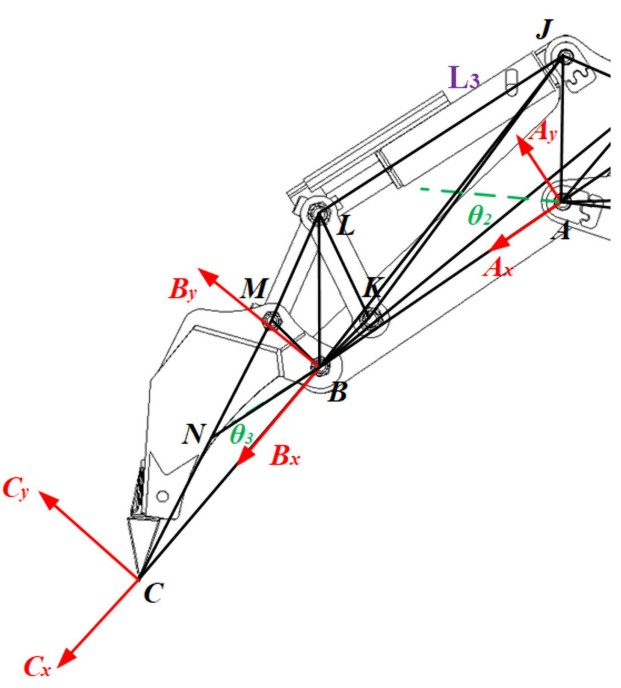

**Fig 5. D-H linkage diagram of the bucket.**

Synthesizing the foregoing derivations, applying the law of cosines to $\Delta MBC$, $\Delta LMB$, $\Delta LJB$ yields, we conclude:

$$\theta_3 = \angle MBC - \angle MBN = \angle MBC + \angle MBL + \angle LBJ + \angle BJA - \pi = \arccos \frac{MB^2 + CB^2 - MC^2}{2MB \times CB} +$$
$$\arccos \frac{MB^2 + LB^2 - LM^2}{2MB \times LB} + \arccos \frac{LB^2 + JB^2 - L_3{}^2}{2LB \times JB} + \arccos \frac{JB^2 + AB^2 - JA^2}{2JB \times AB} - \pi \tag{16}$$

This study analyzes the kinematic relationships among joint variables, linkages, and hydraulic inputs within each circuit of the backhoe loader. By integrating structural parameters of the linkage mechanism with hydraulic inputs, the pose (position and orientation) of all nodal points within the maximum working range is determined. Based on the geometric specifications of the prototype design, the working stroke ranges for each hydraulic cylinder are specified in Table 2.

The joint rotation angle must be calculated in conjunction with the fixed parameters of the links, these parameters are listed in Table 3.

During excavation and ripping operations of the backhoe loader, forward kinematics analysis with hydraulic stroke inputs yields the following joint angle ranges: $\theta_1$: −13.02° to 37.13°; $\theta_2$: 16.95° to 126.42°; $\theta_3$: −34.14° to 55.45°.

**D. Analytical determination of the workspace.**

i. Maximum digging depth

The maximum digging depth ($D_{max}$) of the backhoe loader was computed via kinematic analysis subject to hydraulic system constraints. The configuration achieving $D_{max}$ is shown in Fig 6.

As shown in Fig 6, at the configuration corresponding to the maximum digging depth ($D_{max}$), the working device is at its down-dig limit: the boom hydraulic cylinder is fully retracted and joint angle $\theta_1$ reaches its maximum value (counterclockwise positive). At this point, the stick cylinder aligns the stick parallel to the slewing-ring axis, and the bucket cylinder adjusts the tooth tip $C$ to the position where points $A$, $B$, and $C$ are collinear. This forms a geometric configuration where the stick-bucket assembly, pivoting at point $C$, remains parallel to the slewing ring axis.

Geometric considerations yield:

$$D_{max} = OA \times \sin \theta_{1\,max} + AC - h_1 \tag{17}$$

**Table 2. Drive cylinder operating range.**

| Name of Cylinder | Elongation/(mm) | Contraction/(mm) | Stroke/(mm) |
|---|---|---|---|
| Boom Cylinder ($L_1$) | 780 | 560 | 220 |
| Dipper stick Cylinder ($L_2$) | 1342 | 822 | 520 |
| Bucket Cylinder ($L_3$) | 780 | 520 | 260 |

**Table 3. Fixed parameters for working device links.**

| Name of the Parameter | DO | EO | FO | EF | GO |
|---|---|---|---|---|---|
| Value (mm) | 290.52 | 819 | 804 | 156 | 189.74 |
| Name of the Parameter | OA | GA | IA | IB | BA |
| Value (mm) | 1199.8 | 1102.89 | 318.57 | 972.18 | 661.1 |
| Name of the Parameter | KJ | LK | JB | JA | BK |
| Value (mm) | 737.27 | 264 | 891.3 | 326.69 | 156 |
| Name of the Parameter | LM | MB | MC | CB | |
| Value (mm) | 264 | 149.83 | 659.95 | 634.9 | |

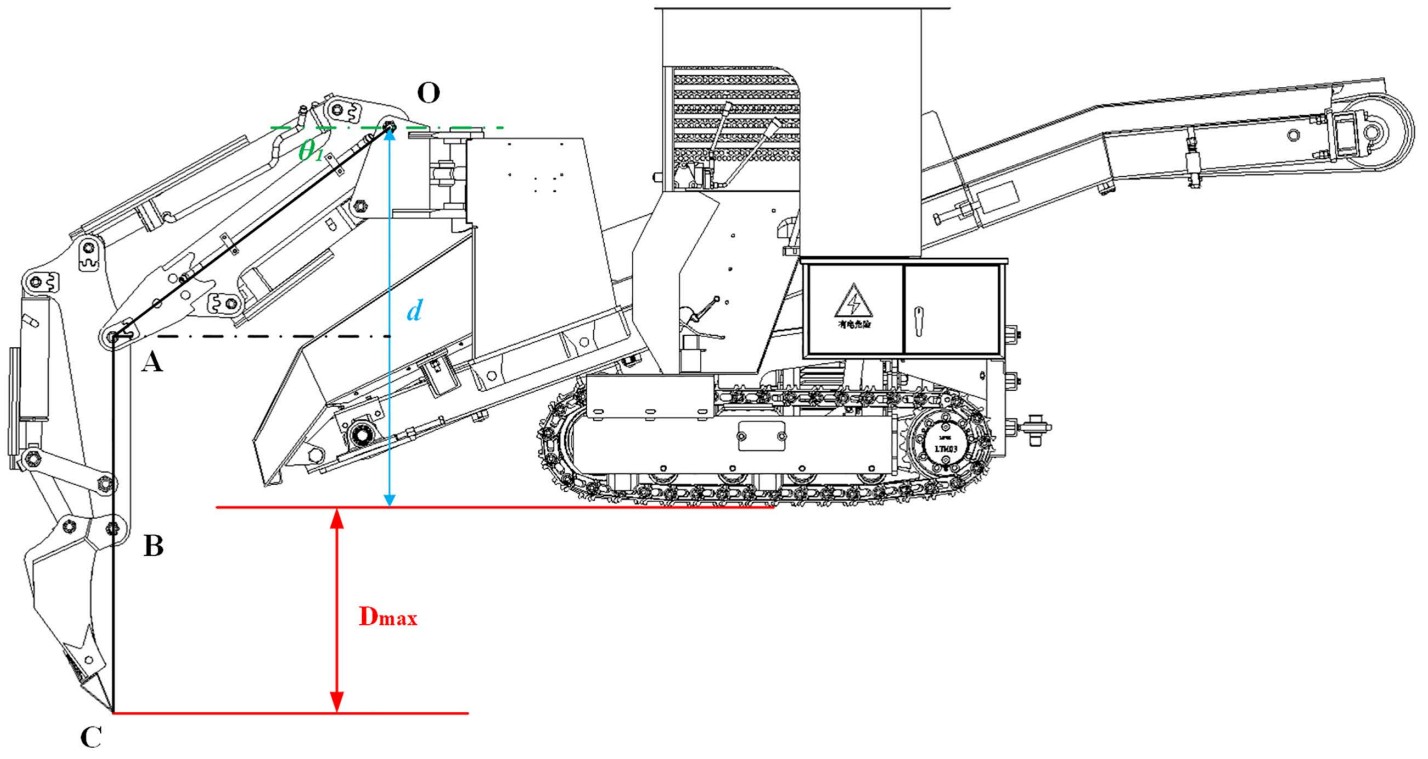

**Fig 6. Configuration at maximum digging depth.**

where $h_1$ is the vertical distance from pivot $O$ (at the boom–slewing ring connection) to the lower surface of the undercarriage crawler track.

Substituting the measured values $h_1 = 1224$ mm and $AC = 1296$ mm, together with the known parameters $OA = 1199.8$ mm and $\theta_{1max} = 37.13°$, into Equation (17) yields $D_{max} = 796.23$ mm.

ii. **Maximum digging height**

The maximum digging height ($H_{max}$) characterizes the highest position attainable by the excavation working device. The configuration achieving $H_{max}$ is shown in Fig 7.

As shown in Fig 7, at the configuration corresponding to the maximum digging height ($H_{max}$), the boom hydraulic cylinder is at full stroke and joint angle $\theta_1$ reaches its minimum value (counterclockwise positive). At the same time, the stick and bucket hydraulic cylinders are at their minimum strokes, with $\theta_2$ and $\theta_3$ simultaneously at their minimum values. From this pose, together with the geometric relations, we obtain:

$$H_{max} = h_1 + h_2 - h_3 + h_4 = h_1 + OA \times \sin\left(|\theta_{1\,min}|\right) - BA \times \left(\sin\left(\theta_{2\,min} - |\theta_{1\,min}|\right)\right) + CB \times \sin\left(|\theta_{3\,min}|\right) \tag{18}$$

Where $h_2$ is the vertical (ground-normal) projection of the line segment connecting pivots $O$ and A; $h_3$ is the vertical (ground-normal) projection of the line segment connecting pivots $A$ and $B$; and $h_4$ is the vertical projection of the line segment connecting the stick–bucket pivot B and the bucket tooth tip C.

Based on kinematic analysis and structural parameters, substituting the known parameters $h_1 = 1224$ mm, $OA = 1199.8$ mm, $BA = 661.1$ mm, $CB = 634.9$ mm, $\theta_{1min} = -13.02°$, $\theta_{2min} = 16.95°$, and $\theta_{3min} = -34.14°$ into Equation (18) yields $H_{max} = 1805.31$ mm.

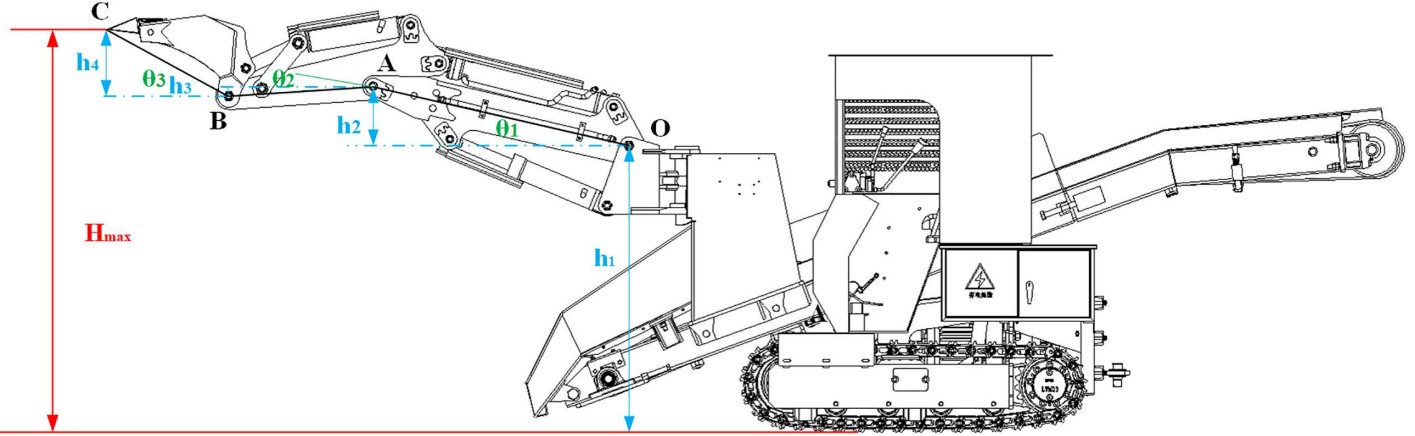

**Fig 7. Configuration at maximum digging height.**

### iii. Maximum digging radius

The maximum digging radius ($R_{max}$) defines the excavator's horizontal working envelope. The configuration achieving $R_{max}$ is shown in Fig 8.

As shown in Fig 8, at the configuration corresponding to the maximum digging radius, the boom hydraulic cylinder orients the boom parallel to the slewing-ring base, while the stick and bucket hydraulic cylinders remain at their minimum strokes. In this configuration, $\theta_2$ reaches its minimum value and the bucket tooth tip is coincident with the slewing-ring base, yielding the maximum horizontal extension of the working device. From this pose and the geometric relations, we obtain:

$$R_{max} = r_1 + r_2 + r_3 + r_4 = r_1 + r_2 + BA \times \cos\theta_{2\,min} + CB \times \cos\left(\theta_{2\,min} - |\theta_3|\right) \tag{19}$$

where $r_1$ is the perpendicular distance from pivot $O$ to the slewing-ring rotation axis; $r_2$ is the horizontal (plan-view) projection of $OA$ (since the boom is parallel to the slewing-ring base, $r_2 = OA$); $r_3$ is the horizontal projection of $AB$; and $r_4$ is the horizontal projection of $BC$.

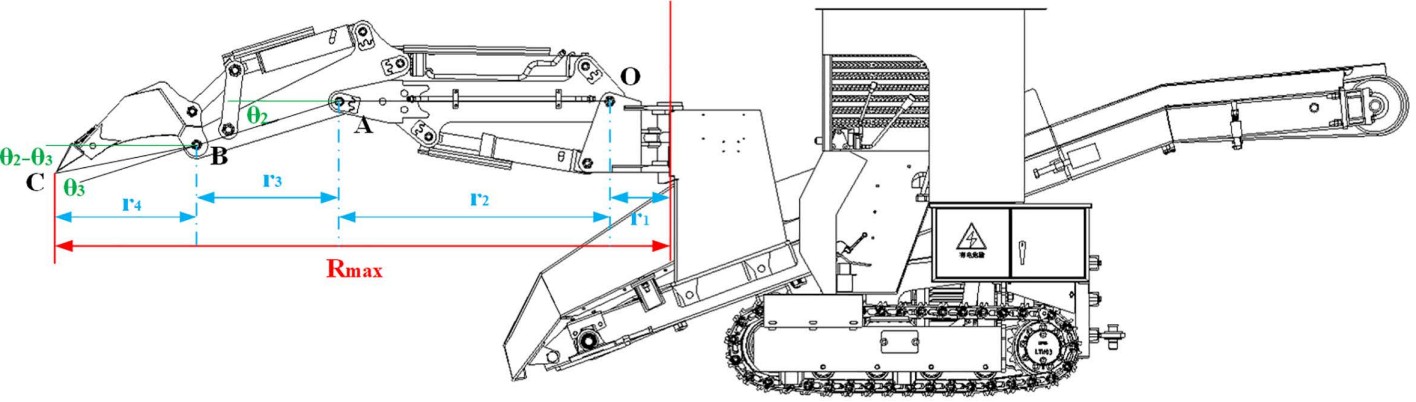

**Fig 8. Configuration at maximum digging radius.**

Based on the preceding kinematic analysis and the working device's geometric and structural parameters, the maximum digging radius ($R_{max}$) was obtained. The key parameters are $r_1 = 265$ mm, $OA = 1199.8$ mm, $BA = 661.1$ mm, $CB = 634.9$ mm, and $\theta_{2min} = 16.95°$. At this configuration, $\theta_3$ was measured as $6.02°$. Substituting these values into Equation (18) yields $R_{max} = 2720.56$ mm.

**E. Multibody dynamics modeling in ADAMS.** During field operations, the working cycle comprises four phases: penetration, excavation, ripping, and conveying. This study focuses on the backhoe loader's performance during the first three phases. Given the wide variability in material properties and associated force dynamics, traditional physical load testing proves time-intensive and costly. Virtual prototyping enables rapid acquisition of hinge joint load data, significantly enhancing efficiency while reducing experimental resource expenditure.

The geometric model was imported into ADAMS for multibody dynamics simulation, the prototype mechanism is driven by prescribed kinematic joints. During simulation, ADAMS evaluates the system's degrees of freedom; redundant degrees of freedom or overconstraints degrade the accuracy of the dynamic solution. Therefore, such redundancies should be removed before conducting dynamic analyses.

To mitigate motion interference during initial assembly, joints with higher mobility (higher DOF) but equivalent functionality are assigned according to the kinematic relationships among the links. Redundant degrees of freedom and overconstraints are then identified using the ADAMS Model Check. Guided by the diagnostics, the flagged connections are replaced with lower-mobility joints to remove redundancies, and the procedure is iterated until the model passes validation. Essential kinematic joints were applied at critical hinge locations (Fig 9). The constraint topology is defined in Table 4.

The backhoe loader performs material excavation via three sets of hydraulic cylinders actuating the boom, dipper arm, and bucket, while the slewing support's bilateral hydraulic systems enable steering operations. Based on the operating parameters of each cylinder listed in Table 2 and the planar working condition considered in this study, step drive functions are designed for the hydraulic cylinders. A rigid-body dynamics model is then constructed and validated, from which the no-load planar working range diagram was obtained (see Fig 10).

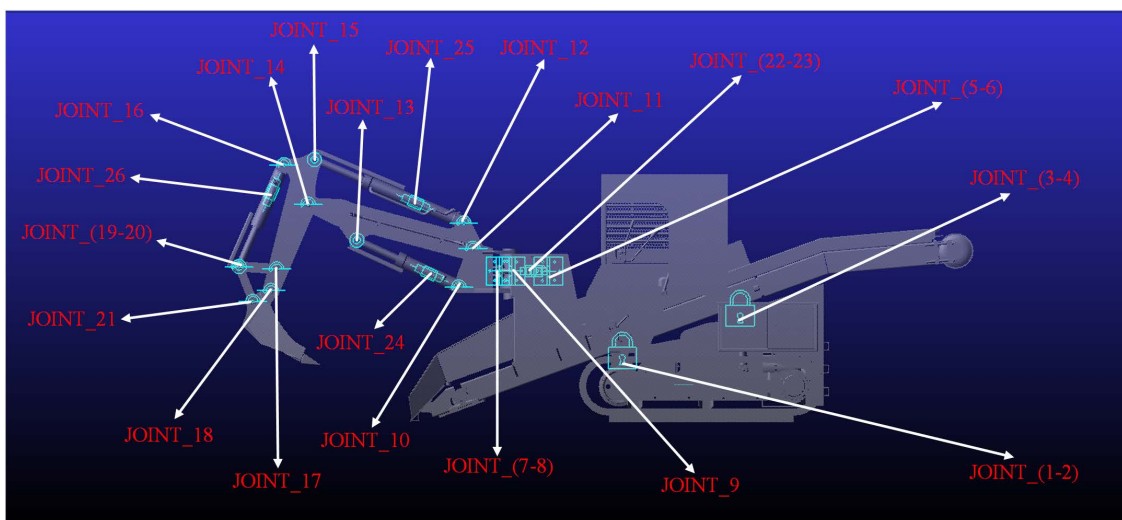

**Fig 9. Constraint diagram of hinge point movement of backhoe loader.**

**Table 4. Reaming point movement constraint attribute of backhoe loader.**

| Rod 1 | Rod 2 | Joint hinge point | Kinematic joint |
|---|---|---|---|
| Undercarriage | Conveying system | JOINT_1 | Fixed joint |
| Undercarriage | Cab | JOINT_2 | Fixed joint |
| Undercarriage | Hydraulic control unit | JOINT_3 | Fixed joint |
| Gantry | Conveying system | JOINT_4 | Fixed joint |
| Slewing left cylinder barrel | Gantry | JOINT_5 | Revolute Joint |
| Slewing right cylinder barrel | Gantry | JOINT_6 | Revolute Joint |
| Slewing support | Gantry | JOINT_7 | Revolute Joint |
| Slewing left cylinder rod | Slewing support | JOINT_8 | Revolute Joint |
| Slewing right cylinder rod | Slewing support | JOINT_9 | Revolute Joint |
| Boom cylinder barrel | Slewing support | JOINT_10 | Revolute Joint |
| Boom | Slewing support | JOINT_11 | Revolute Joint |
| Dipper stick cylinder barrel | Boom | JOINT_12 | Revolute Joint |
| Boom cylinder barrel | Boom | JOINT_13 | Cylindrical Joint |
| Dipper stick | Boom | JOINT_14 | Revolute Joint |
| Dipper stick cylinder rod | Dipper stick | JOINT_15 | Cylindrical Joint |
| Bucket cylinder barrel | Dipper stick | JOINT_16 | Revolute Joint |
| Bucket rocker | Dipper stick | JOINT_17 | Revolute Joint |
| Bucket actuator | Dipper stick | JOINT_18 | Revolute Joint |
| Bucket cylinder rod | Rocker | JOINT_19 | Cylindrical Joint |
| Actuator | Rocker | JOINT_20 | Revolute Joint |
| Bucket | Actuator | JOINT_21 | Revolute Joint |
| Slewing left cylinder rod | Slewing left cylinder barrel | JOINT_22 | Translational joint |
| Slewing right cylinder rod | Slewing right cylinder barrel | JOINT_23 | Translational joint |
| Boom cylinder rod | Boom cylinder barrel | JOINT_24 | Translational joint |
| Dipper stick cylinder rod | Dipper stick cylinder barrel | JOINT_25 | Translational joint |
| Boom cylinder rod | Boom cylinder barrel | JOINT_26 | Translational joint |

In the global coordinate system, negative X-direction displacement of the bucket tooth tip denotes extension (Fig. 11(a)). The bucket reaches the maximum digging radius when the tooth-tip X displacement attains its minimum. In the simulation, the tooth-tip X displacement at t = 5.925s is 9507.9 mm, which corresponds to the configuration at the maximum digging radius.

At JOINT_9 (the pivot between the slewing base and the gantry frame), a marker was placed to track displacement along the X axis, and the results are presented in Fig 12.

During operation, the marker at JOINT_9 remained stationary, so its X position was fixed at 12,190 mm. Using the previously defined criterion for the maximum digging radius, the X-axis separation between JOINT_9 and the bucket tooth tip was computed, yielding $R_{max}$ = 2682.1 mm under the forward digging condition.

As shown in Fig. 11(b), the bucket tooth-tip displacement along the Y axis is reported in the global frame, with extension defined as the + Y direction. The configuration of maximum digging height is attained when the tooth-tip displacement y(t) reaches its maximum. In the present simulation, at t = 3.99 s the tooth-tip Y displacement reaches an extremum of $y_{max}$ = −728.1 mm.

At the bottom of the undercarriage, a marker was placed to track Y-direction displacement, and the resulting time history is presented in Fig 13.

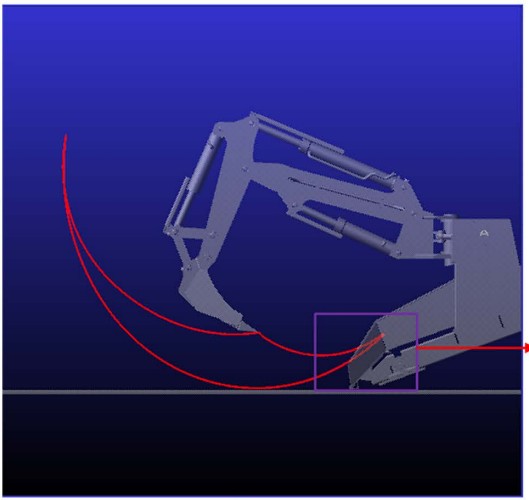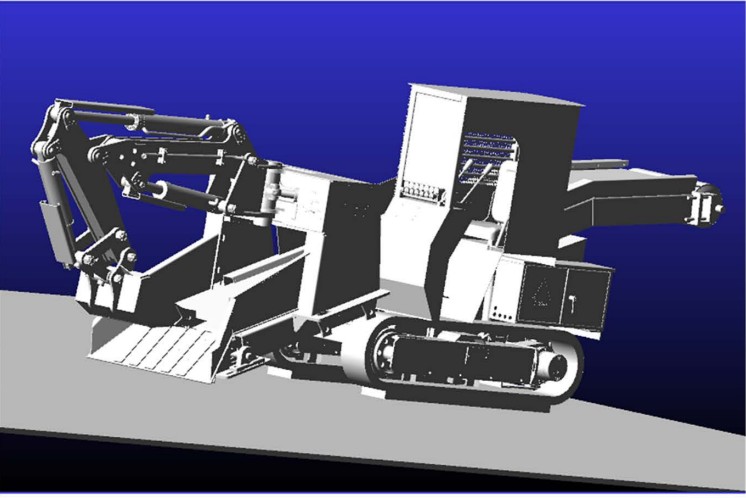

**Fig 10. Working range diagram.** The planar (X–Y) trajectory obtained from the bucket tooth-tip marker captures the time-varying motion characteristics of the tooth tip, as shown in Fig 11. (a) X-direction displacement of the bucket tooth tip. (b) Y-direction displacement of the bucket tooth tip.

During operation, the reference marker at the undercarriage bottom remains stationary at $y = -2500$ mm in the global frame. According to the previously defined criterion for the maximum digging height, the Y-direction difference between the bucket tooth-tip position and this reference ($H_{max} = y_{marker} - y_{tooth}$) yields $H_{max} = 1771.9$ mm under the forward-digging condition.

According to the ADAMS simulation results, under the forward-digging configuration on level ground (without reaching the maximum digging-depth posture), the obtained maximum digging height and maximum digging radius were compared with the operating parameters from the previous forward-kinematics analysis. The comparison results are shown in Table 5.

Comparison between the kinematic analysis and ADAMS simulations yields relative errors of 1.41% for the maximum digging radius and 1.85% for the maximum digging height, confirming the accuracy of the prototype model and the reliability of the kinematic analysis.

## F. Pose analysis based on hydraulic-cylinder parameters

Based on the dynamic analysis of the excavation condition in this study, the displacement and velocity time histories for the three hydraulic cylinders of the prototype are shown in Figs 14-16.

Analysis of the displacement and velocity time histories of hydraulic cylinder indicates that the boom hydraulic cylinder reaches full stroke and remains stationary (zero extension/retraction velocity) during 2–5 s, whereas the stick and bucket cylinders reach full stroke and stabilize during 4–5 s. Accordingly, 0–4 s corresponds to the working device lifting phase, and at $t = 4$ s the working device attains its maximum digging height; the corresponding configuration is shown in Fig. 17.

After $t = 5$ s, the working device enters the lowering phase. As shown in Fig 16(b), the bucket-cylinder velocity in the Y direction attains an extremum at $t = 8.95$ s, indicating completion of the lowering motion and transition to the configuration of maximum digging depth (Fig. 18). At this point, the operating mode shifts from digging to back-dragging and conveying, corroborating the depth limit and the associated change in the operating envelope.

As a joint examination of the velocity time histories for the boom, stick, and bucket cylinders (Fig. 14(b), Fig. 15(b), Fig. 16(b)) shows, at $t = 10.975$ s the three profiles intersect with a concurrent sign reversal, marking the transition from the back-dragging and conveying mode to the dumping/reset phase (Fig. 19).

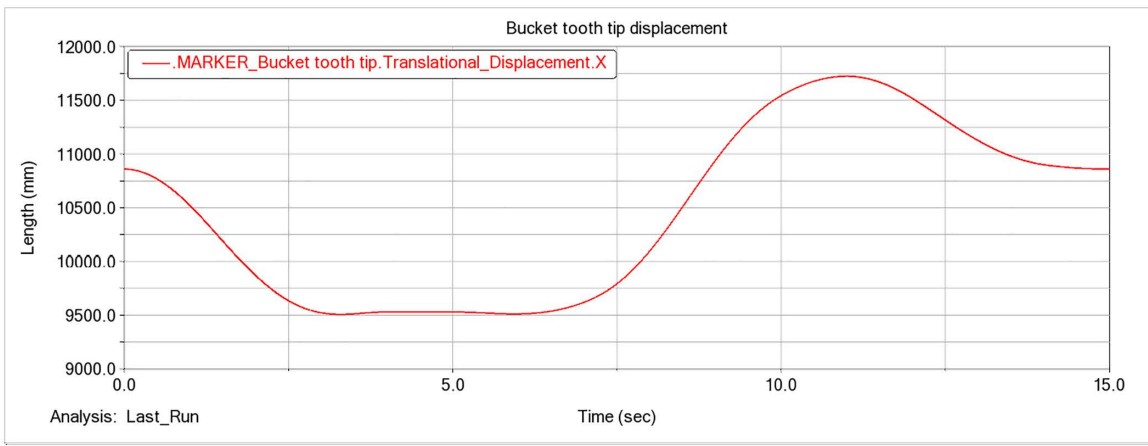

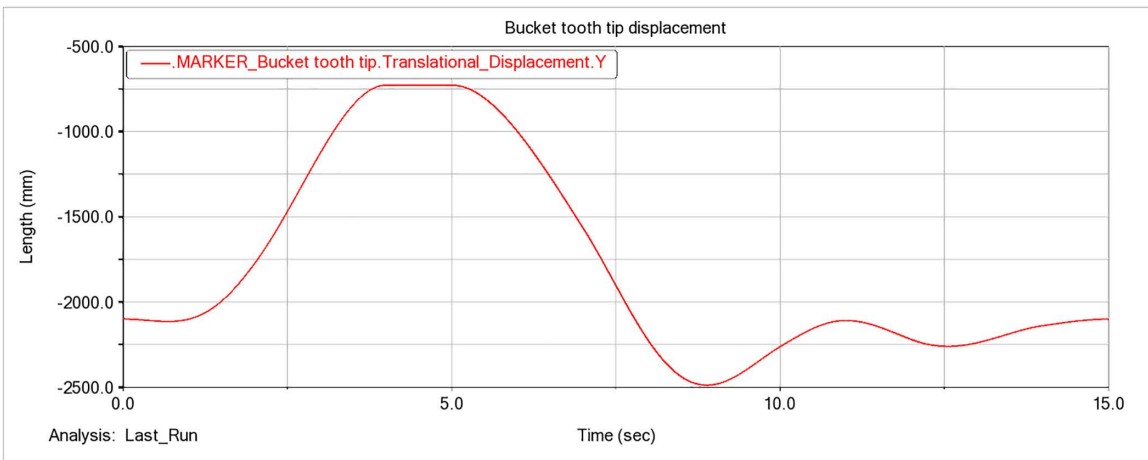

**Fig 11. Displacement variation of the bucket tooth tip.**

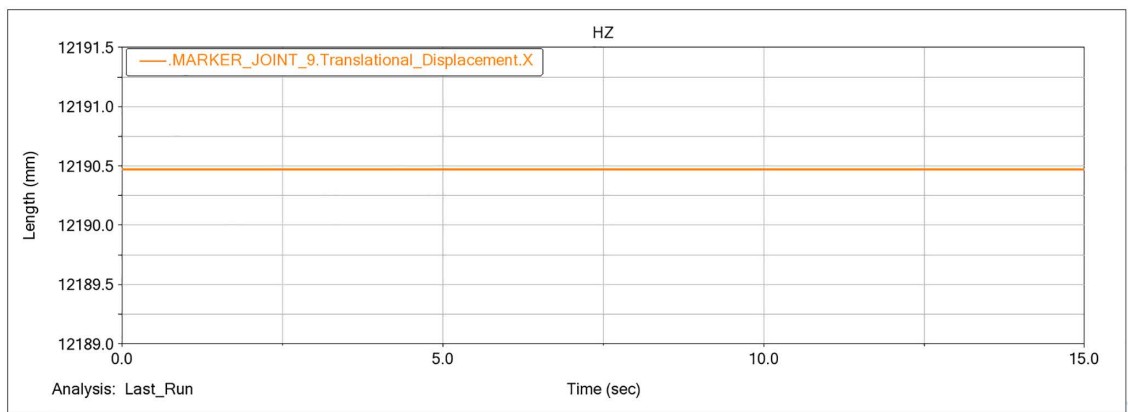

**Fig 12. X.direction displacement time history of the marker at JOINT_9.**

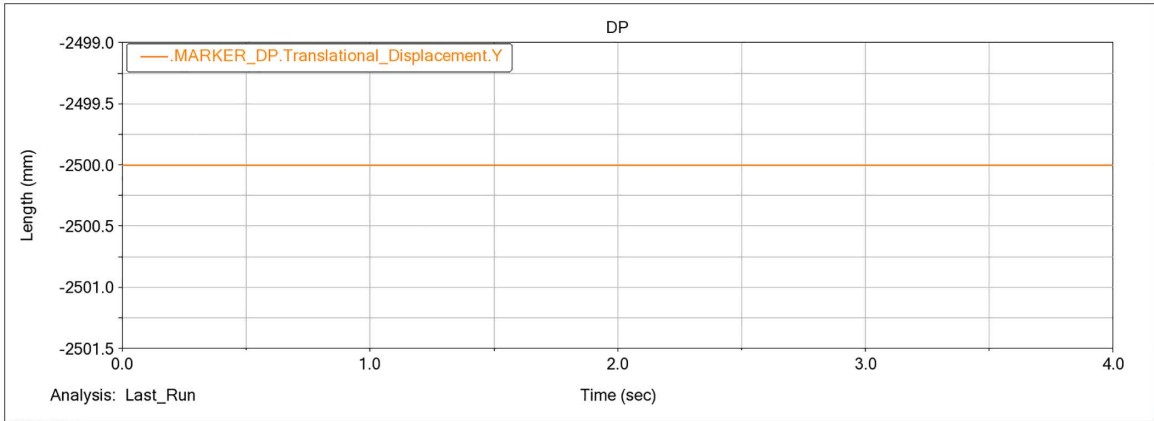

**Fig 13. Y-direction displacement time history of the marker at the undercarriage bottom.**

**Table 5. Comparison of operating parameters.**

| Operating Parameters | Results from kinematic analysis | Results from ADAMS simulation | Relative error |
|---|---|---|---|
| Maximum digging radius | 2720.56 mm | 2682.1 mm | 1.41% |
| Maximum digging height | 1805.31 mm | 1771.9 mm | 1.85% |

## III. Development and operating condition analysis of an EDEM-coupled model

**A. Model development in EDEM.** Validation against experimental data reported in the literature [16–17] showed small discrepancies between load spectra from discrete element method–multibody dynamics (DEM–MBD) simulations and measurements, and the simulated motion tracked the observed kinematics. The Enhanced Discrete Element Method (EDEM) was coupled with the multi-body dynamics simulation software ADAMS to effectively analyze the digging operation of the backhoe loader and to monitor in real time the contact forces between the bucket and the material, as well as the load variations at the hinge joints of the connecting rods. During the construction of the EDEM model, fundamental parameters for both the bulk particles and structural components—such as density, Poisson's ratio, shear modulus, and Young's modulus—must be defined. In this study, ore particles and Q235 structural steel, commonly used in engineering applications, were selected as representative materials in Table 6.

The contact parameters between materials—such as the static friction coefficient, rolling friction coefficient, and coefficient of restitution—must be determined, typically through physical or virtual experiments. According to Reference [18], the contact parameters between the backhoe loader and ore particles in the EDEM model are listed in Table 7.

Simulating ore materials using a single particle type may result in significant deviations from real-world behavior. To enhance the accuracy of the material excavation simulation, a set of spherical particles with varying diameters was employed to generate a more diverse particle assembly. Based on the simplification of five representative material shapes, Reference [19] proposed five particle models—ellipsoidal, triangular pyramidal, cylindrical, cubic, and prismatic (see Fig 20)—which were used to complete the excavation simulation (see Fig 21). Given the diversity of particle morphologies in real operating conditions, cylindrical and ellipsoidal particles each constitute a relatively small fraction. Prior studies [20] report that the mass ratio of cylindrical particles to all other shapes is approximately 0.7:1, and the ratio of ellipsoidal particles to all other shapes is likewise approximately 0.7:1. In line with engineering practice that further down-weights slender morphologies, we therefore set each of these ratios to 0.65:1 in this study [21]. These five particle shapes were employed to construct the stockpile distribution model. Detailed information on the quantity and proportion of each particle type is provided in Table 8.

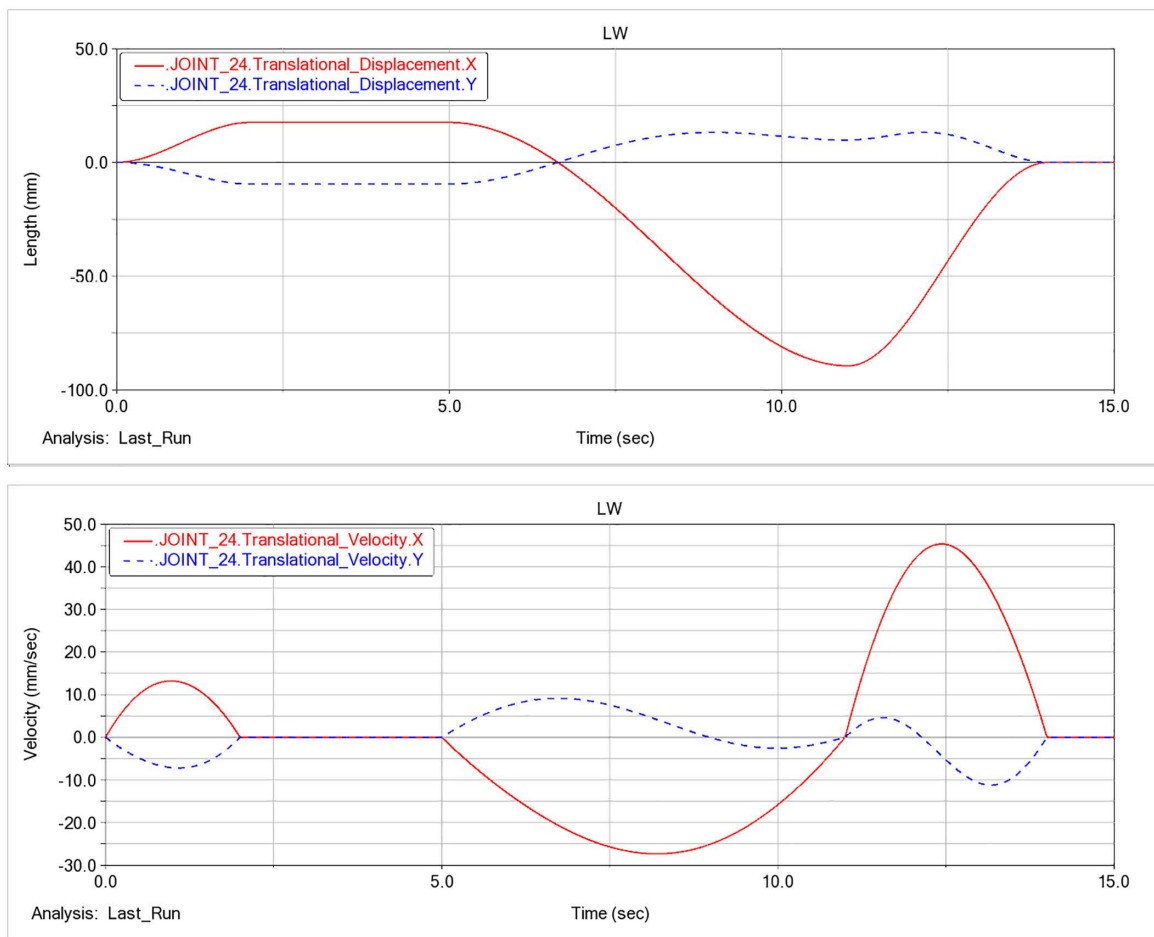

**Fig 14. Displacement and velocity time histories of the boom hydraulic cylinder.** (a) Displacement time history of the boom hydraulic cylinder. (b) Velocity time history of the boom hydraulic cylinder.

**B. Loading condition analysis for the backhoe loader.** A coupled EDEM–ADAMS simulation was conducted to evaluate two typical operating conditions: front digging and side digging. The corresponding digging resistance curves of the bucket under both scenarios are presented in Fig 22.

As shown in Fig 23, the peak digging resistance for both operation modes is concentrated during the excavation phase, accompanied by sudden fluctuations. These abrupt load changes subject the structural members of the backhoe loader to extreme working conditions, highlighting the need for structural optimization to ensure stable excavation performance.

To identify the critical working conditions and corresponding load information, the hinge point loads listed in Table 4 were extracted using the Adams post-processing module. In this study, the boom of the backhoe loader was selected as the analysis target, and the hinge load data under both operational scenarios are presented.

The simulation results indicate that, during the 15-second operation cycle, the hinge point loads under forward excavation exhibit two distinct peaks: one at 7.65 s (onset of digging) and another at 8.9 s (onset of material delivery), which correspond to the peak moments of the digging resistance observed in Fig 23(a).

In contrast, the hinge load distribution under lateral excavation is more dispersed. A comprehensive analysis in conjunction with the excavation resistance curve shown in Fig 23(b) reveals a localized peak at 7.96 s, occurring in the initial stage of excavation.

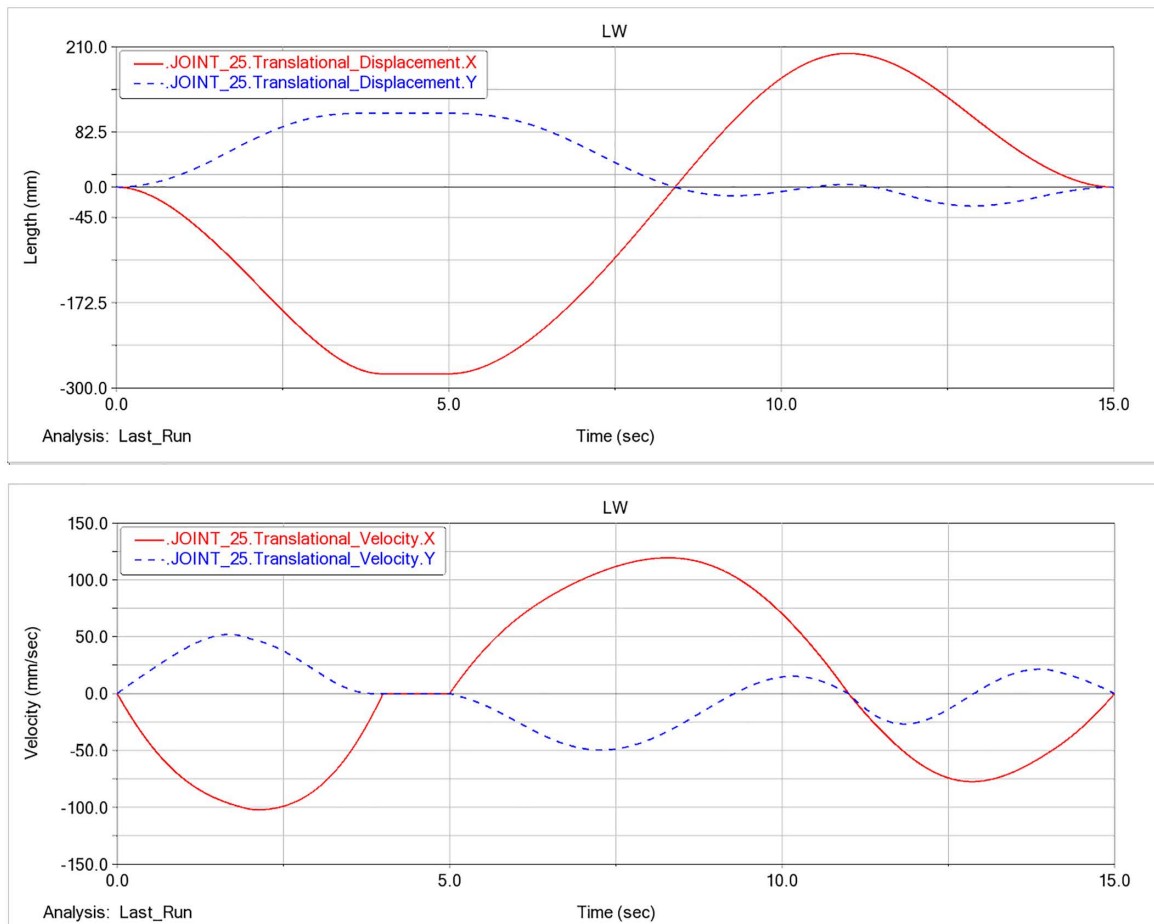

**Fig 15. Displacement and velocity time histories of the stick hydraulic cylinder. (a) Displacement time history of the stick hydraulic cylinder. (b) Velocity time history of the stick hydraulic cylinder.**

Fig 24 shows that the peak average velocity of material movement under forward excavation occurs at 7.7 s, covering both peak load phases at 7.65 s and 8.9 s. In contrast, the peak velocity under lateral excavation appears at 9 s, which aligns with its corresponding peak load phase.

Based on a comprehensive analysis of excavation resistance, hinge point loads, and material movement velocity, the critical working moments for forward excavation are identified as the initial excavation stage (7.65 s) and the onset of material delivery (8.9 s). For lateral excavation, the critical moment is determined to be the initial excavation stage (7.96 s). The corresponding system postures at these moments are illustrated in Fig 25.

**C. Static analysis and identification of extreme working conditions for the boom.** Based on the EDEM–ADAMS co-simulation results, the force data for each hinge point of the boom under both operating modes were obtained, as summarized in Table 9.

Based on the stress data at the hinge points provided in the table, the model was imported into ANSYS for the following preprocessing and setup steps:

i. Material Property Definition.

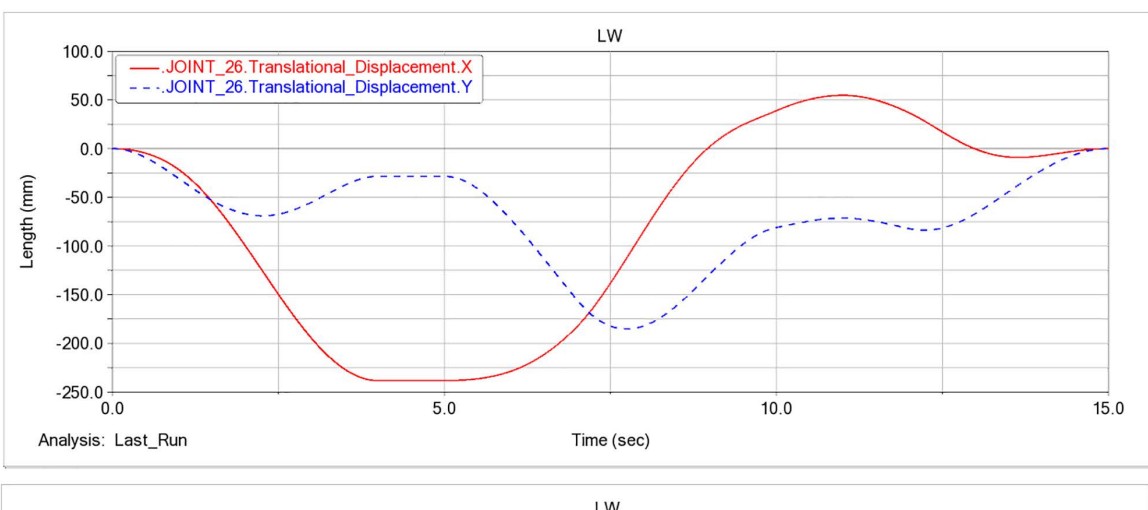

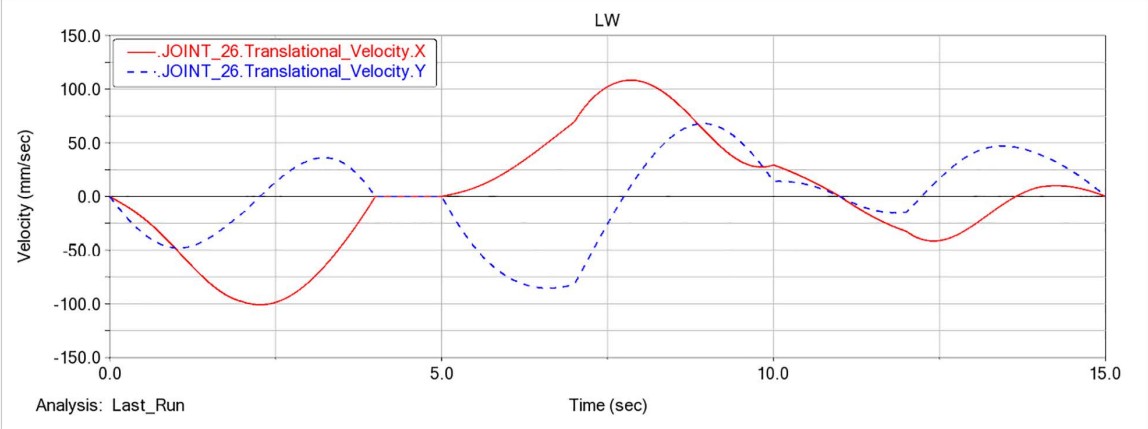

**Fig 16. Displacement and velocity time histories of the bucket hydraulic cylinder.** (a) Displacement time history of the bucket hydraulic cylinder. (b) Velocity time history of the bucket hydraulic cylinder.

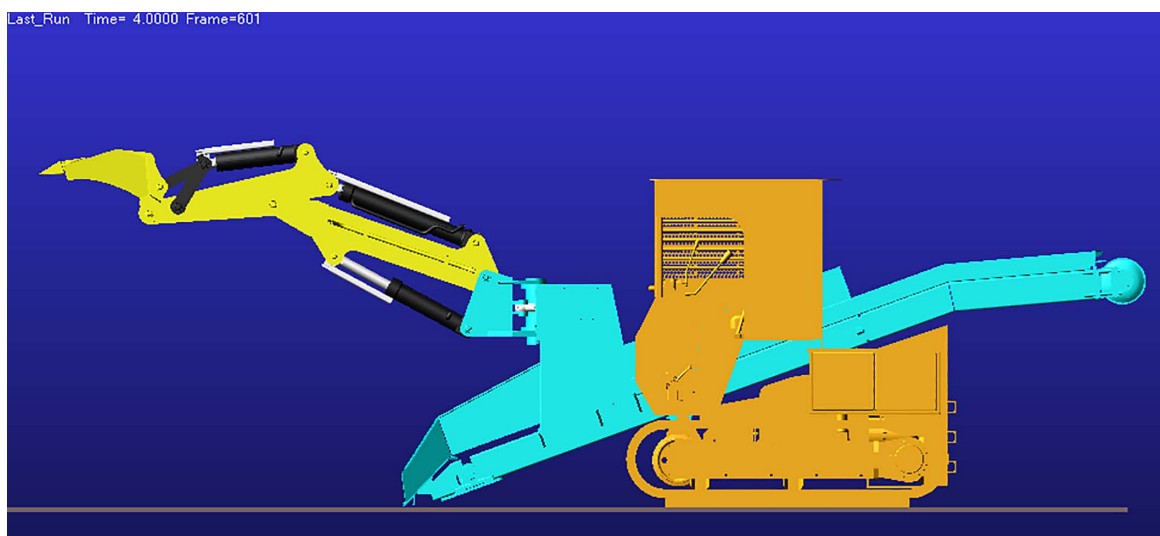

**Fig 17. Configuration at maximum digging height.**

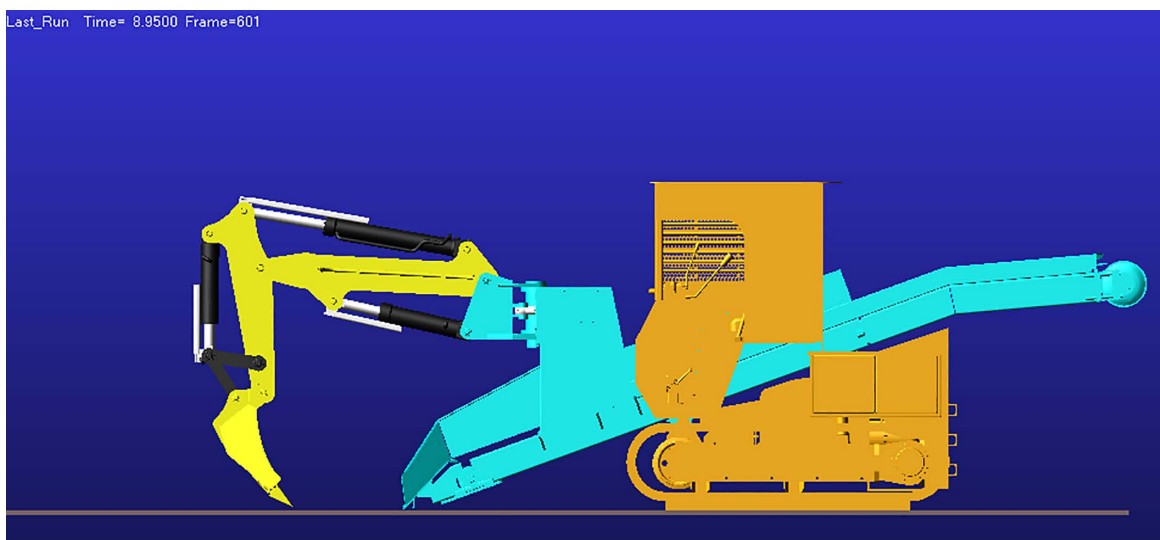

**Fig 18. Configuration at maximum digging depth.**

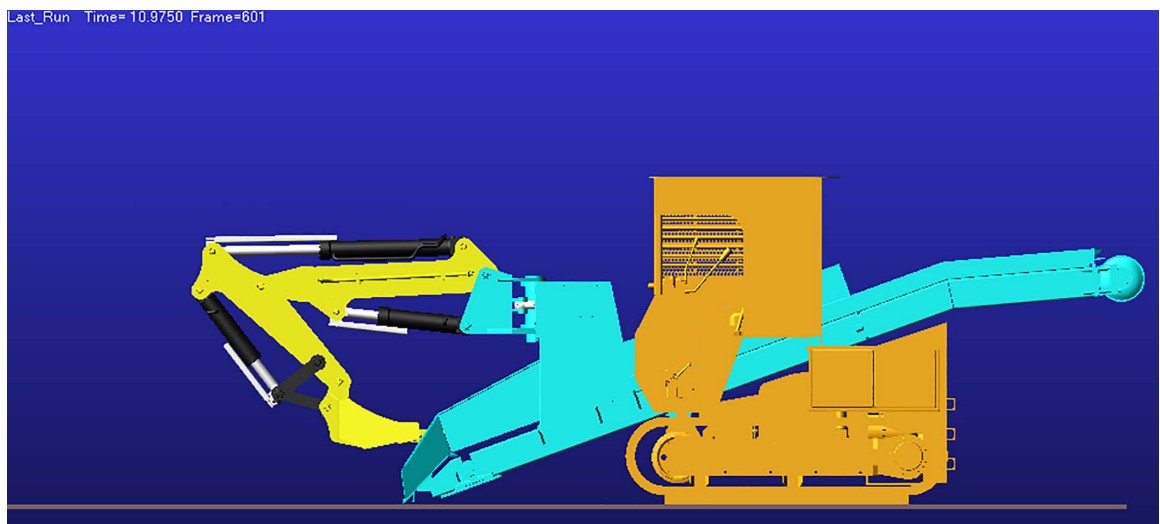

**Fig 19. Configuration at the dumping/reset phase.**

**Table 6. Material intrinsic attribute parameters.**

| Material type | Poisson's ratio | modulus (Pa) | Density (kg/m³) |
|---|---|---|---|
| Ores | 0.3 | $1 \times 10^7$ (Shear modulus) | 1600 |
| Q235 steel | 0.29 | $2.07 \times 10^{11}$ (Young's modulus) | 7801 |

**Table 7. EDEM model contact parameters.**

| Material \ Coefficient | Coefficient of restitution | Static friction coefficient | Kinetic friction coefficient |
|---|---|---|---|
| Inter-particle interactions (of ores) | 0.3 | 0.5 | 0.15 |
| Between the particle and the structural arm | 0.5 | 0.4 | 0.04 |

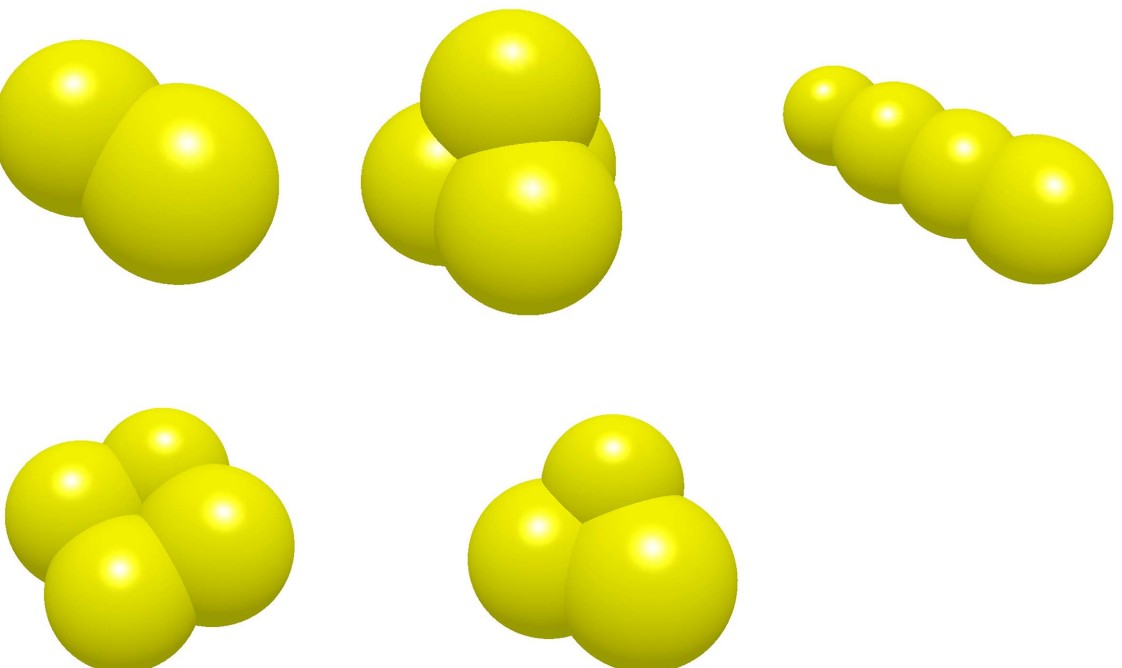

**Fig 20. Modeling of particle unit.** (a) Ellipsoidal (b) Triangular pyramid (c) Cylindrical(d) Cubic (e) Triangular prism.

Time: 0 s

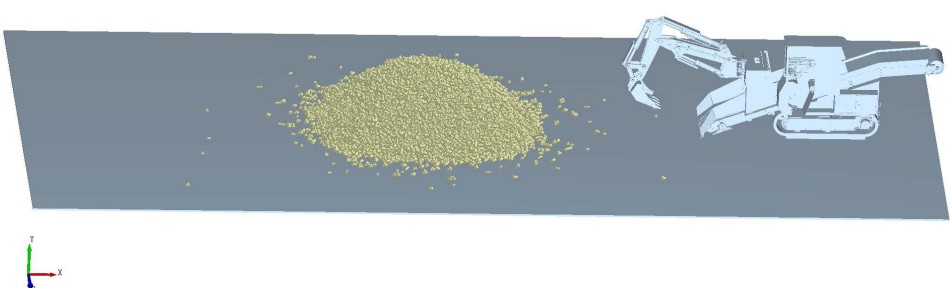

**Fig 21. EDEM-ADAMS coupled modeling.**

**Table 8. The composition proportion of five types of particles.**

| Particle type | Particle mass (kg) | Proportion (%) | Total mass (kg) |
|---|---|---|---|
| Ellipsoidal | 0.044 kg-M1 | 30 | 780 |
| | 1.3*M1 | 30 | |
| | 1.6*M1 | 25 | |
| | 2*M1 | 15 | |
| Triangular pyramid | 0.068 kg-M2 | 30 | 1200 |
| | 1.3*M2 | 30 | |
| | 1.6*M2 | 25 | |
| | 2*M2 | 15 | |
| Cylindrical | 0.074 kg-M3 | 30 | 780 |
| | 1.3*M3 | 30 | |
| | 1.6*M3 | 25 | |
| | 2*M3 | 15 | |
| Cubic | 0.067 kg-M4 | 30 | 1200 |
| | 1.3*M4 | 30 | |
| | 1.6*M4 | 25 | |
| | 2*M4 | 15 | |
| Triangular prism | 0.053 kg-M5 | 30 | 1200 |
| | 1.3*M5 | 30 | |
| | 1.6*M5 | 25 | |
| | 2*M5 | 15 | |

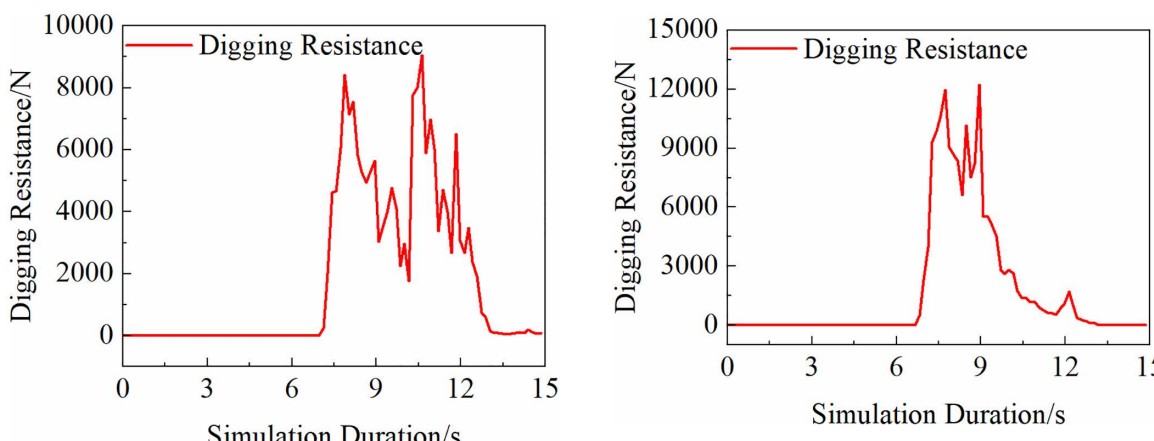

**Fig 22. Excavation resistance diagram.** (a) Digging resistance under forward excavation conditions. (b) Digging resistance under lateral excavation.

Considering that the bucket is in direct contact with the ground and hard materials during operation and is subjected to high contact loads, Q235 structural steel was selected as the material. Its key mechanical properties are listed in Table 10.

ii. Meshing.

Regular tetrahedral elements were generated using ANSYS's free meshing algorithm. To enhance simulation accuracy, a local mesh refinement was applied at the hinge regions with a cell size of 5 mm, while a 10 mm mesh size was used for the rest of the model.

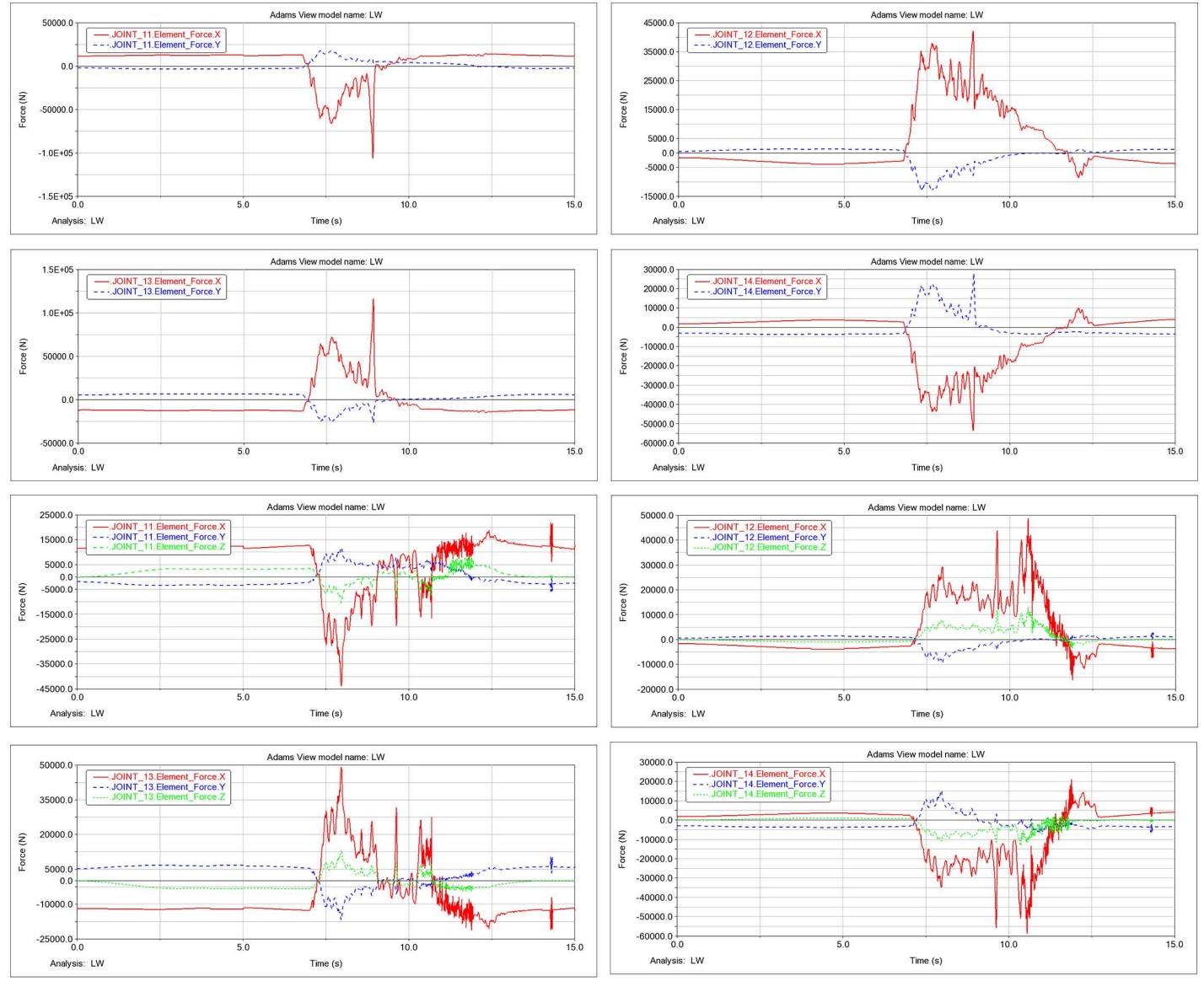

**Fig 23. Stress curve of hinge point of boom under two working conditions.** (a) Force Curve of JOINT_11 under Forward Excavation Conditions. (b) Force Curve of JOINT_12 under Forward Excavation Conditions. (c) Force Curve of JOINT_13 under Forward Excavation Conditions. (d) Force Curve of JOINT_14 under Forward Excavation Conditions. (e) Force Curve of JOINT_11 under Lateral Excavation Conditions. (f) Force Curve of JOINT_12 under Lateral Excavation Conditions. (g) Force Curve of JOINT_13 under Lateral Excavation Conditions. (h) Force Curve of JOINT_14 under Lateral Excavation Conditions.

### iii. Load Setup.

Combining the operating postures at each time step in the EDEM-ADAMS coupled simulation with the hydraulic cylinder's working range as shown in Table 1, the working posture of each linkage under the extreme loading condition was determined. Corresponding force parameters from Table 8 were applied to the model. Fixed constraints were also introduced, as illustrated in Fig 26.

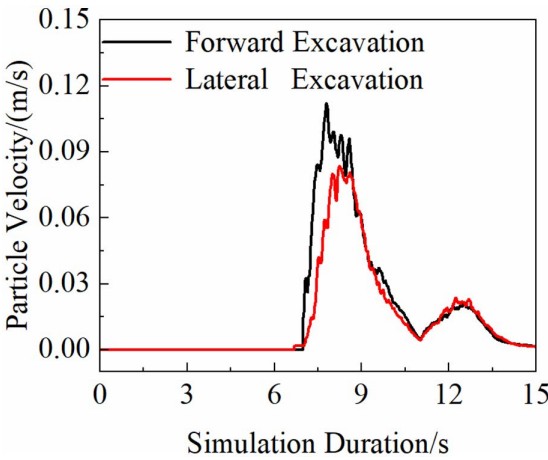

**Fig 24. Material movement speed.**

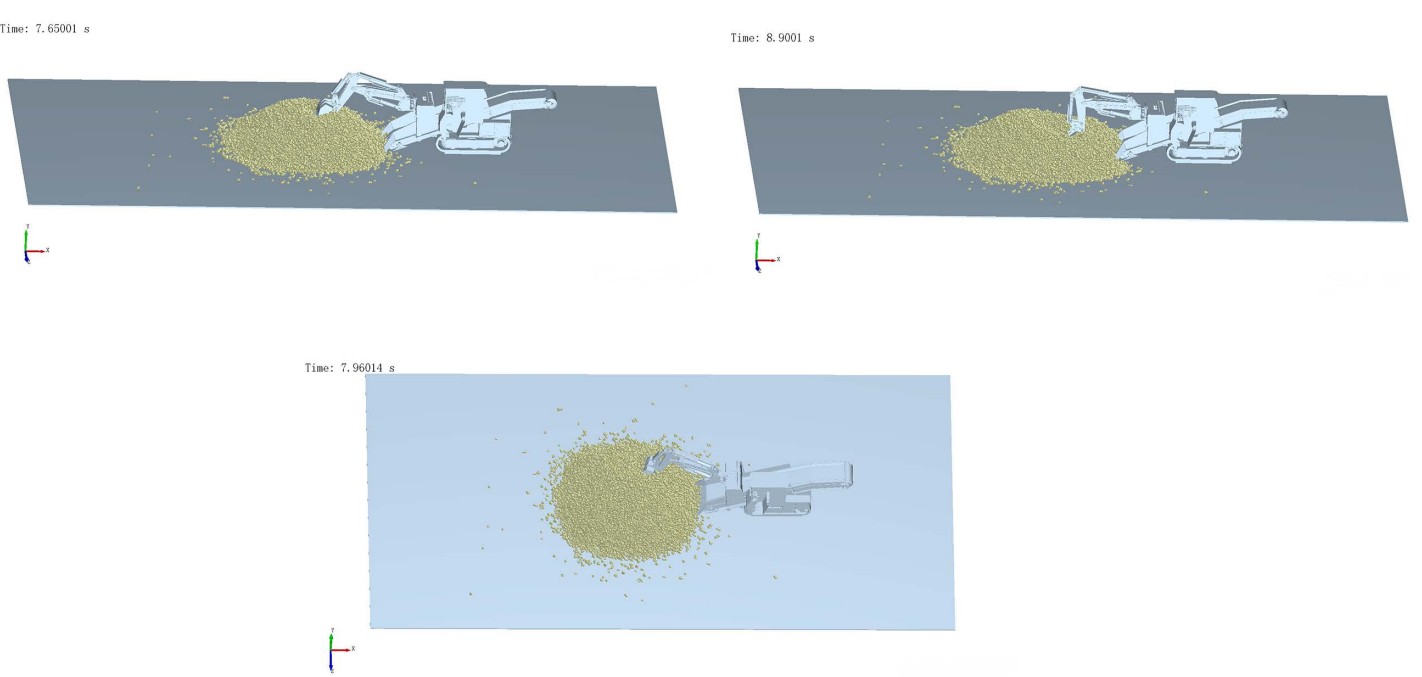

**Fig 25. Peak working state of side excavation.** (a) Working Posture of the Forward Excavation Mechanism at 7.65 s. (b) Working Posture of the Forward Excavation Mechanism at 8.9 s. (c) Working Posture of the Lateral Excavation Mechanism at 7.96 s.

After completing the above simulation setup, the equivalent stress contours and total deformation distributions of the boom under each working condition were obtained, as shown in Fig 27.

The results of the finite element analysis (FEA) reveal significant differences in the mechanical response of the boom under three sets of extreme loading conditions.

Notably, the forward excavation condition at $t = 8.9$ s represents the most critical scenario, exhibiting a peak equivalent stress of 131.42 MPa and a maximum deformation of 0.10777 mm. In contrast, the lateral excavation condition at $t = 7.96$ s yields the lowest mechanical response, with a peak equivalent stress of 104.13 MPa and a deformation of 0.072722 mm.

**Table 9. Force parameters of each hinge point in each working condition of boom.**

| Excavation condition | Boom hinge | $F_x$(N) | $F_y$(N) | $F_z$(N) |
|---|---|---|---|---|
| 7.65 s (Forward Digging) | JOINT_11 | −66183 | 17549 | 0 |
| | JOINT_12 | 37850 | −12934 | 0 |
| | JOINT_13 | 71970 | −26000 | 0 |
| | JOINT_14 | −43636 | 22445 | 0 |
| 8.9 s (Forward Digging) | JOINT_11 | −93751 | 8500 | 0 |
| | JOINT_12 | 41791 | −7794 | 0 |
| | JOINT_13 | 105380 | −24396 | 0 |
| | JOINT_14 | −53417 | 24753 | 0 |
| 7.96 s (Lateral Digging) | JOINT_11 | −43890 | 11863 | −10102 |
| | JOINT_12 | 29253 | −9364 | 7838 |
| | JOINT_13 | 49068 | −16719 | 13148 |
| | JOINT_14 | −34430 | 15283 | −10884 |

**Table 10. Q235 Material properties.**

| Material | Modulus of elasticity/(MPa) | Poisson's ratio | Density/(kg/m³) | Yield strength/(MPa) | Tensile strength/(MPa) |
|---|---|---|---|---|---|
| Q235 steel | 2.06*10⁵ | 0.28 | 7850 | 235 | 390~620 |

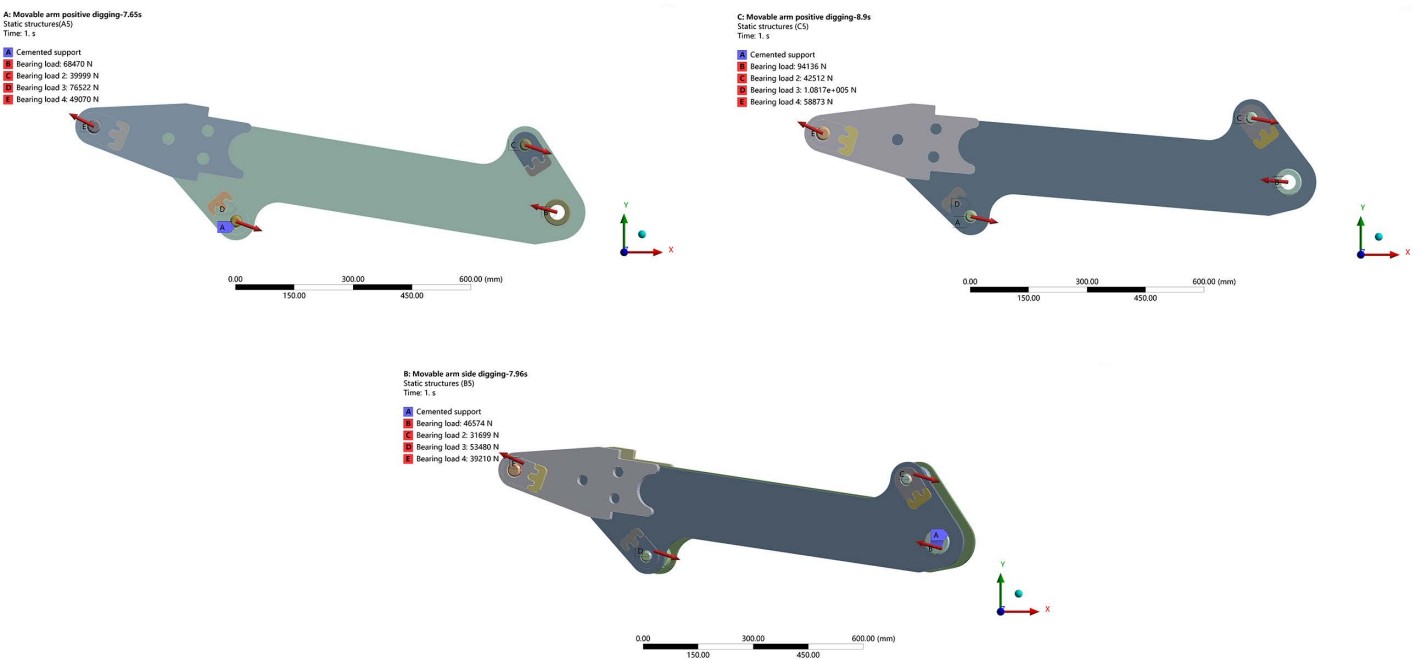

**Fig 26. Boom load setting.** (a) Load Application for Boom at 7.65 s under Forward Excavation. (b) Load Application for Boom at 8.9 s under Forward Excavation. (c) Load Application for the Boom at 7.96 s under Lateral Excavation.

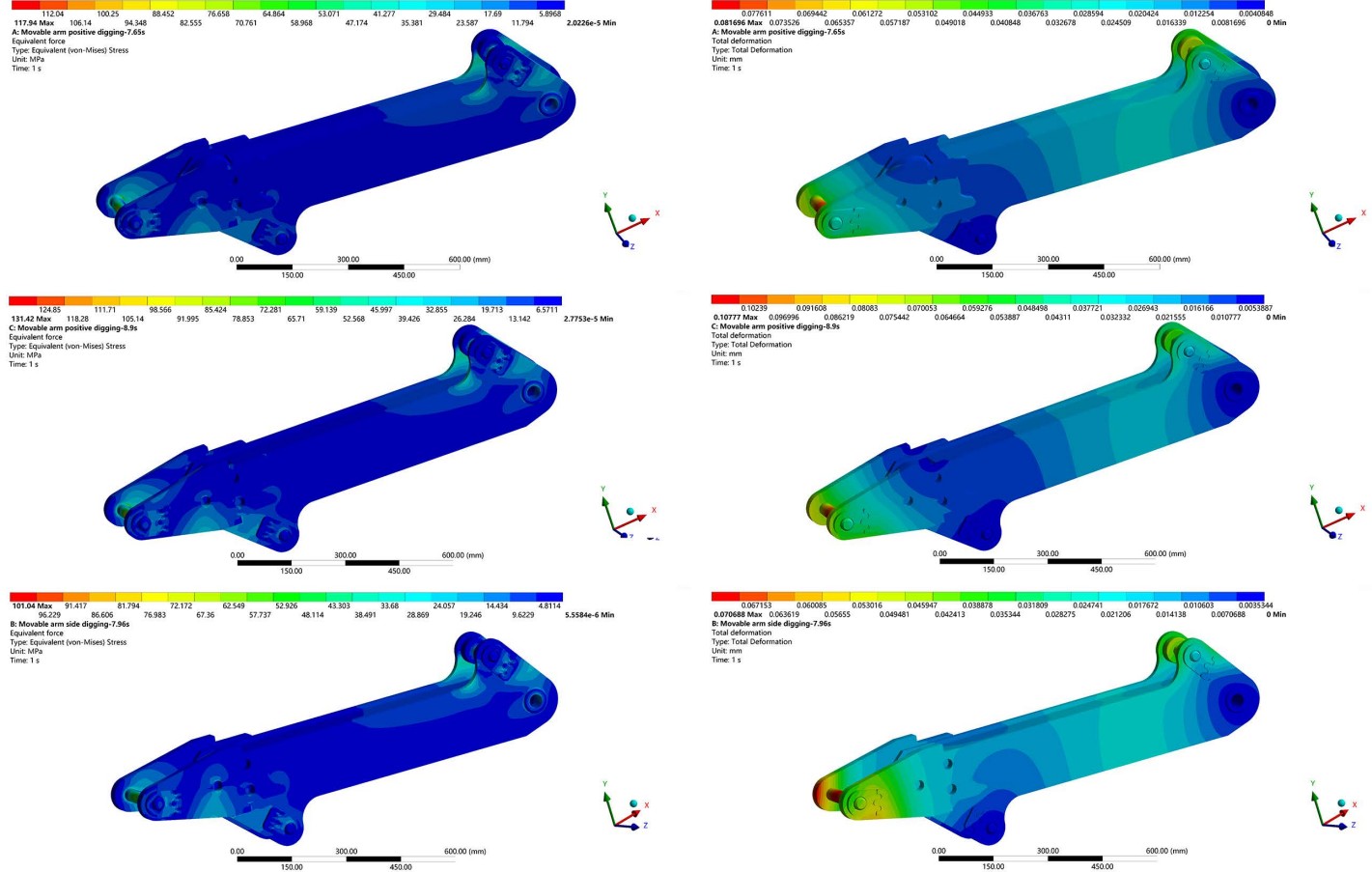

**Fig 27. Finite element analysis cloud picture of each limit working condition of boom.** (a) Equivalent Stress Contour of the Boom at 7.65 s under Forward Excavation. (b) Total Deformation Contour of the Boom at 7.65 s under Forward Excavation. (c) Equivalent Stress Contour of the Boom at 8.9 s under Forward Excavation. (d) Total Deformation Contour of the Boom at 8.9 s under Forward Excavation. (e) Equivalent Stress Contour of the Boom at 7.96 s under Lateral Excavation. (f) Total Deformation Contour of the Boom at 7.96 s under Lateral Excavation.

As shown in Fig 28, when the boom–gantry system is in the horizontal position, the boom cylinder and dipper (bucket rod) cylinder form a parallel kinematic pair, while the bucket cylinder maintains a vertical orientation. This kinematic configuration corresponds spatially to the maximum digging depth condition illustrated in Fig 18.

Based on this correspondence, it can be concluded that the initial stage of material delivery—corresponding to the maximum working depth posture—constitutes the most critical structural condition. At this moment, the bucket volumetric efficiency reaches its peak, and the corresponding velocity field distribution of material flow is shown in Fig 29.

Finite element analysis of the boom across three ultimate-load cases showed that, under the forward-excavation condition, both the equivalent stress and the total deformation peaked at 8.9 s.

Therefore, 8.9 s under forward excavation is identified as the critical working condition for the backhoe loader, and subsequent optimization of the boom design will focus on this time point.

## IV. Design optimization of the boom and comparative validation

**A. Design optimization of the boom.** To address the structural optimization requirements of the boom and bucket rod, this study employs topology optimization to achieve improved performance and lightweight design. Topology

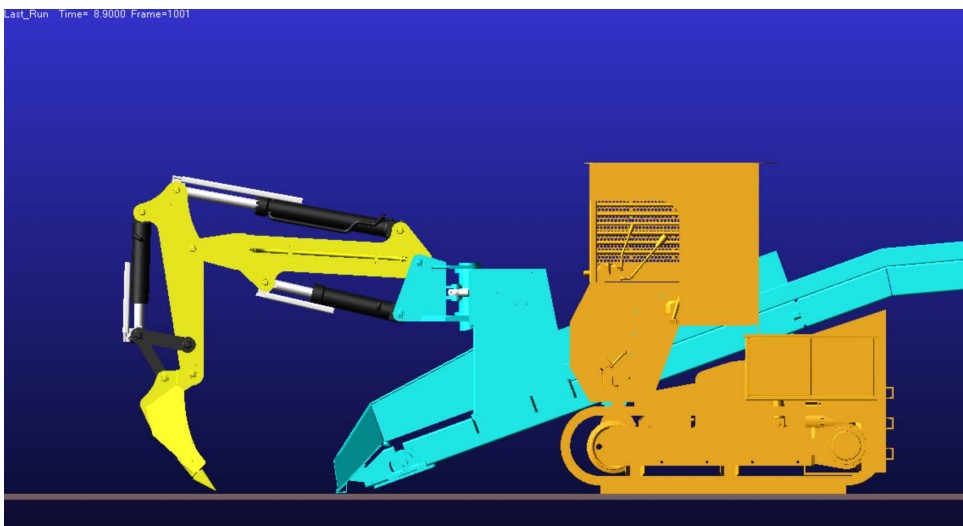

**Fig 28. Attitude of boom in extreme working condition.**

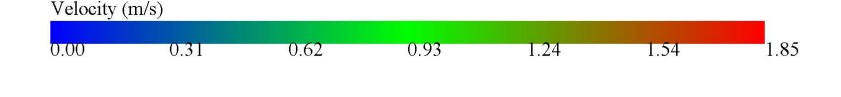

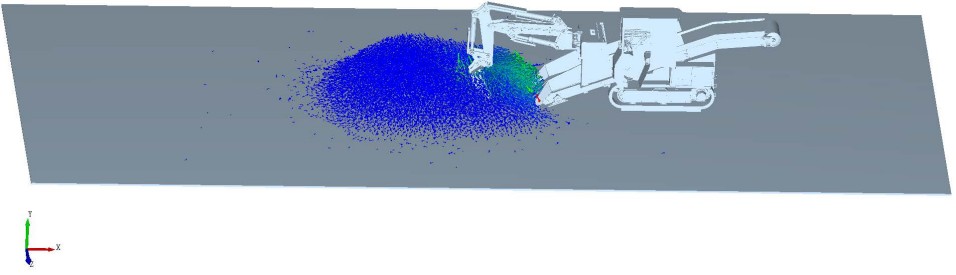

**Fig 29. Direction and speed of material movement under extreme condition of boom.**

optimization reformulates the structural design task as a material distribution problem, and its mathematical formulation is as follows:

$$\begin{cases} \text{Minimize} f(x) \\ \text{Subject to} g_i(x) \leq 0, i = 1, 2, ..., m \\ h_j(x) \leq 0, j = 1, 2, ..., n \\ x_{min} \leq x \leq x_{max} \end{cases}$$

(20)

Where $x$ is the design variable, $f(x)$ is the objective function, and $g_i(x)$ and $h_j(x)$ are the constraint functions. Based on the improved SIMP (solid isotropic material with penalization) method [22], the objective function is formulated as:

$$\begin{cases} f(x) = U^T K U = \sum_{e=1}^{N} E_e(x_e) u_e^T k_0 u_e \\ E_e(x_e) = E_{min} + x_e^p (E_0 - E_{min}) \\ KU = F \\ 0 \le x_e \le 1 \end{cases}$$

(21)

Where $x_e$ is the density assigned to element e, and $E_e$ is the Young's modulus determined by $x_e$, $E_0$ is the stiffness of the solid materia, $E_{min}$ is a very small stiffness assigned to void regions in order to prevent the stiffness matrix from becoming singular, and $p$ is the penalization exponent (typically $p=3$), $U$ and $F$ are the global displacement and force vectors, respectively, $K$ is the global stiffness matrix, $u_e$ is the displacement vector of element $e$, $k_0$ is the element stiffness matrix evaluated for unit Young's modulus.

Based on the load conditions derived from static stress analysis, and under the premise of maintaining the integrity of key functional regions, the design objective for the excavation mechanism is to maximize stiffness and minimize deformation. The excavator working device is subjected to extreme, highly time-varying loads. To maintain a high safety factor in complex operating conditions, a conservative mechanism design was adopted on the basis of prior studies [23]. The boundary condition is defined as a mass retention ratio of 70%, and the connection hinge regions are designated as exclusion zones in the optimization.

Material is removed from non-load-bearing regions, and the final topology-optimized structure is obtained through the optimization algorithm, as shown in Fig 30.

Based on the topology optimization results and the equivalent stress distribution of the boom, the structure in the stress concentration regions was reinforced, while the redundant regions were redesigned for weight reduction. The optimized and reconstructed model is shown in Fig 31.

To verify the reliability of the optimized model, a static simulation analysis was conducted on the boom under the same excavation conditions, applying identical loading and boundary conditions. The analysis results are shown in Fig 32.

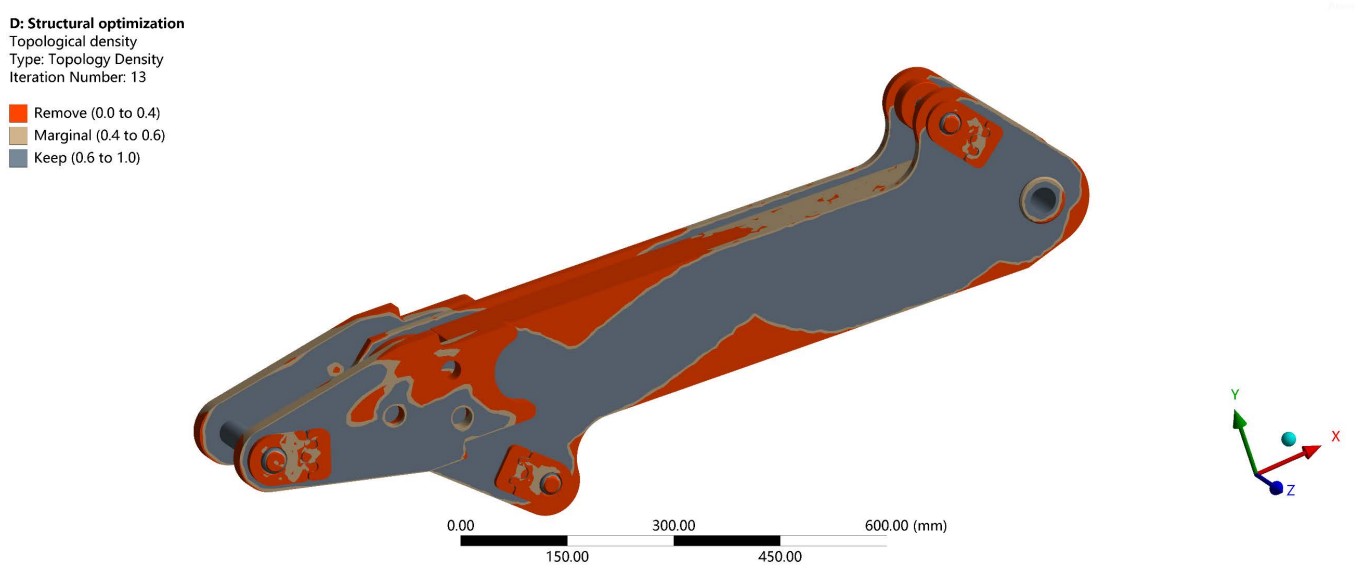

**Fig 30. Load image of boom topology optimization under extreme working conditions.**

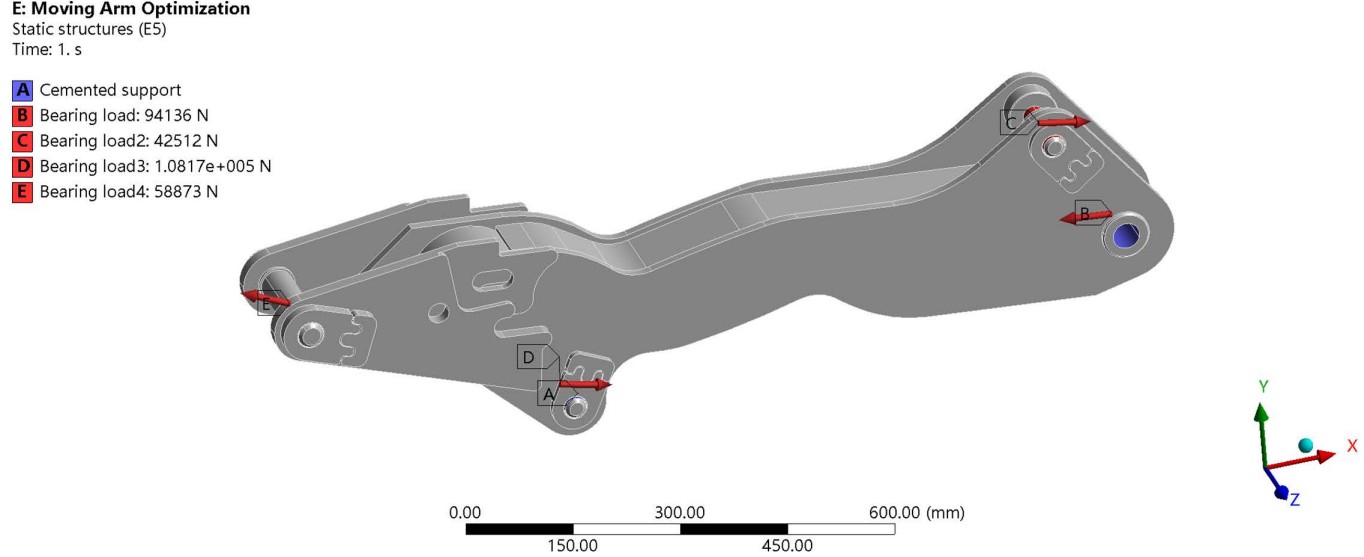

**E: Moving Arm Optimization**
Static structures (E5)
Time: 1. s

**A** Cemented support
**B** Bearing load: 94136 N
**C** Bearing load2: 42512 N
**D** Bearing load3: 1.0817e+005 N
**E** Bearing load4: 58873 N

**Fig 31. Topology optimization after the boom model diagram.**

Based on the topology optimization, the weight of the boom was reduced by 13.80%, and both the equivalent stress and total deformation decreased significantly. These improvements enhance the structural strength and stability, while achieving effective weight reduction. The comparative results before and after optimization are summarized in Table 11.

**B. Comparative validation of optimization under different load spectra.** Different load-spectrum datasets affect the optimization results. In existing load-spectrum acquisition methods, mechanical analysis based on the thrust and torque equilibrium of the hydraulic cylinders is performed to obtain the bucket-tip force, which is then applied to an ADAMS multibody prototype model to derive load-spectrum data at critical hinge points for shape optimization.

In this study, an EDEM–ADAMS simulation is used to reproduce the actual material excavation process, obtain hinge-point load spectra under the most critical operating conditions, and perform optimization analysis of the boom. The results are compared with the structural optimization outcomes obtained using the existing load-spectrum acquisition method [24].

Finite-element analysis results for the boom under the most critical operating condition for the design optimized using the existing load-spectrum acquisition method, are shown in Fig 33.

As shown in Fig 33, under the shape optimization with existing load spectrum acquisition method, the boom exhibits an equivalent stress of 119.52 MPa and a total deformation of 0.098287 mm under the extreme working condition.

In contrast, the optimized model obtained using the proposed method achieves an equivalent stress of 97.10 MPa and a total deformation of 0.076402 mm, representing reductions of 18.76% and 22.27%, respectively.

These results demonstrate a significant improvement in the structural strength and safety of the boom under hazardous loading conditions.

## V. **Conclusions.**

In this study, a kinematic and structural optimization analysis was performed on a small-scale digging transportation device, with a focus on improving the boom's performance under the most critical working condition.

i. Construction of the Kinematic Model: The kinematic model of the backhoe excavator arm is simplified and established using the Denavit–Hartenberg (D-H) method. Forward kinematic analysis is conducted based on the D-H coordinate

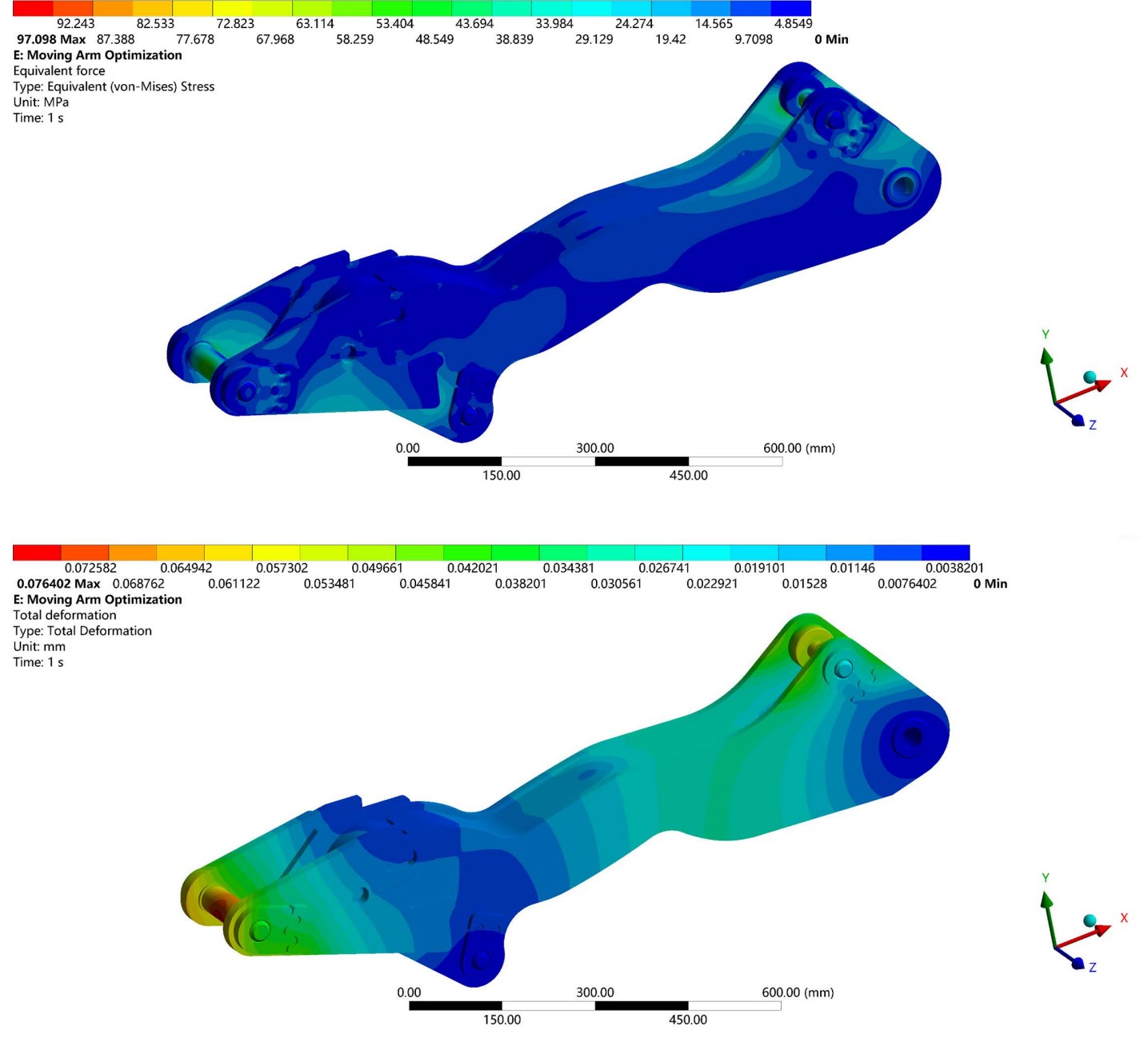

**Fig 32. Optimization model static structure analysis diagram under extreme conditions.** (a) Equivalent Stress Contour of the Optimized Boom. (b) Total Deformation Contour of the Optimized Boom.

system and mechanical parameters, providing a foundation for subsequent dynamic analysis and evaluation of excavation working conditions.

ii. Dynamic simulation and working condition identification: The ADAMS software was used to simulate the system dynamics and to validate the kinematic model through the resulting bucket workspace. Excavation resistance and

**Table 11. Comparison of optimization results.**

|  | Mass/kg | Maximum stress/MPa | Total deformation/mm |
|---|---|---|---|
| Before Optimization | 103.81 | 131.42 | 0.10777 |
| After Optimization | 89.48 | 97.098 | 0.076402 |
| Optimization ratio/% | −13.80% | −26.12% | −29.11% |

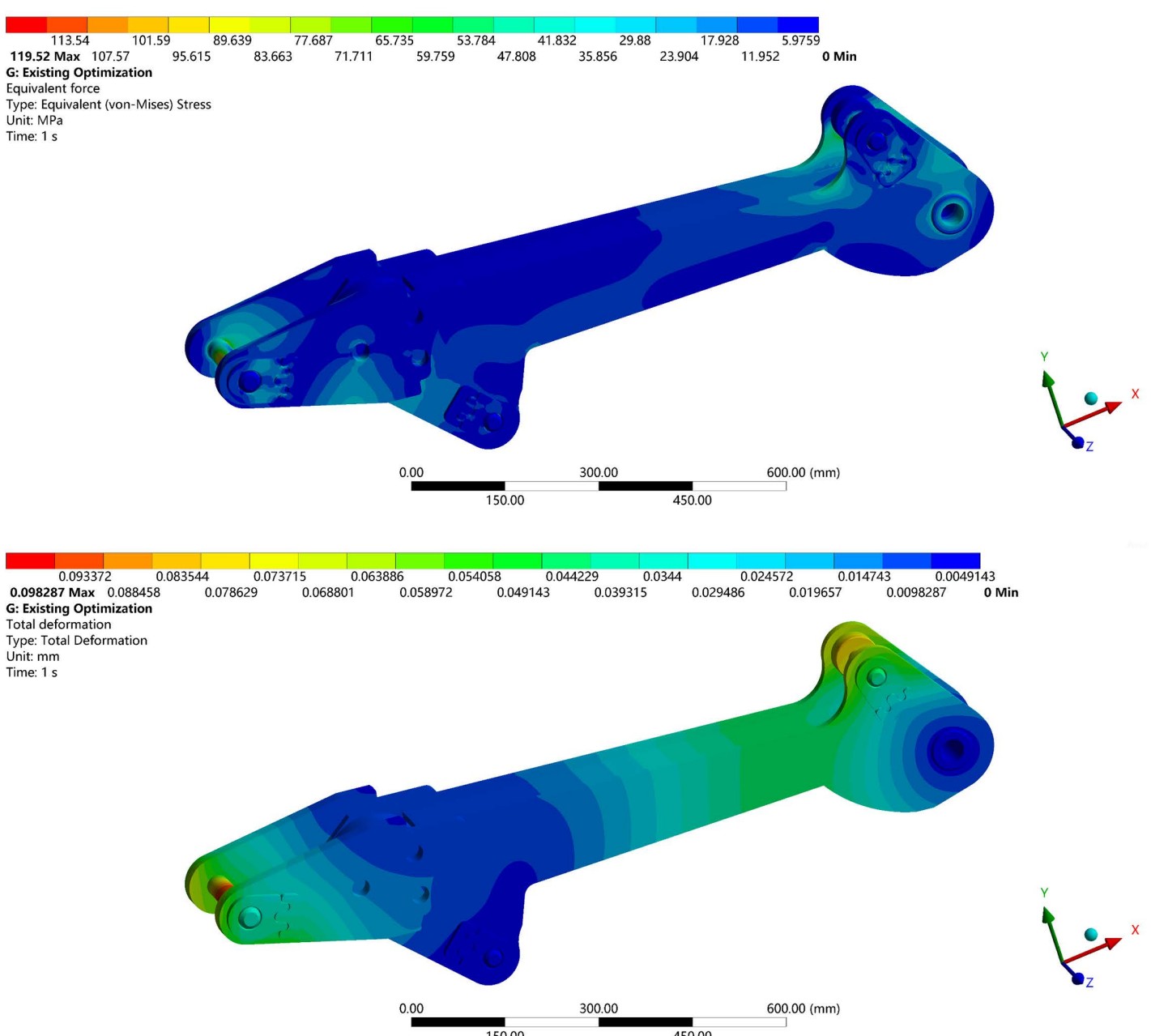

**Fig 33. Static structure analysis diagram of the existing method boom optimization model under extreme conditions.** (a) Equivalent Stress Contour of the Boom Optimized by the Conventional Method. (b) Total Deformation Contour of the Boom Optimized by the Conventional Method.

hinge-point loads were obtained via EDEM-based excavation simulation, allowing identification of the most critical working condition. Static analysis indicated that the most dangerous condition occurs at 8.9 s during forward digging, corresponding to the deepest excavation phase and serving as the basis for structural optimization.

iii. Structural optimization: Through static analysis and topology optimization, the boom's weight was reduced by 14.32 kg (13.80%), the maximum equivalent stress decreased by 26.12%, and the total deformation decreased by 29.11%. Compared with the model optimized by conventional methods, the proposed approach further reduced equivalent stress and deformation by 18.76% and 22.27%, respectively. These improvements not only achieved structural lightweighting but also significantly enhanced operational safety, providing a valuable reference for the design and optimization of future backhoe loader.

## VI. Prospects

This study systematically analyzed the kinematic and dynamic characteristics of the backhoe loader's working device and completed the identification of extreme operating conditions and the structural optimization design. Building on the present results, future work may proceed along the following directions:

i. Multi-field coupled simulation under complex conditions: Extend the scope to diverse geological conditions and operating environments. Employ refined coupled-simulation techniques to comprehensively evaluate the dynamic response of the working device under variable conditions.

ii. Intelligent real-time control: Integrate kinematics, dynamics, and real-time feedback to construct an intelligent operation model, enabling dynamic adjustment to operating conditions and path optimization to improve operational efficiency and environmental adaptability.

This research provides theoretical support for backhoe loader design optimization. Subsequent studies will deepen the current findings, drive technological innovation, and advance the intelligent and efficient development of construction machinery.

## Supporting information

**S1 File. Mechanical data and finite element analysis (FEA) results under the conventional load-spectrum acquisition method.**
(ZIP)

## Author contributions

**Funding acquisition:** Huawei wu.

**Methodology:** Xiaolong Ding, Gui Liu.

**Project administration:** Hualiang Wang.

**Software:** Xiaolong Ding.

**Supervision:** Xiaoyuan Zhu.

**Writing – original draft:** Xiaolong Ding, Gui Liu.

**Writing – review & editing:** Huawei wu, Xiaolong Ding, Gui Liu.

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
