## [Decision Letter · Decision Letter 0]

29 Aug 2025

PONE-D-25-31404Structural Optimization of the Excavator Boom Under Extreme Working Conditions Using EDEM–ADAMS Coupled Simulation Structural optimization designPLOS ONE

Dear Dr. 丁,

Thank you for submitting your manuscript to PLOS ONE. After careful consideration, we feel that it has merit but does not fully meet PLOS ONE’s publication criteria as it currently stands. Therefore, we invite you to submit a revised version of the manuscript that addresses ALL the points raised by reviewers during the review process.

We look forward to receiving your revised manuscript.

Kind regards,

Carlos Alberto Cruz-Villar, Ph. D.

Academic Editor

PLOS ONE

Journal Requirements:

5. Please amend the manuscript submission data (via Edit Submission) to include author Gui Liu, Xiaoyuan Zhu and Hualiang Wang.

6. We note that Figure 9,13 and 17 in your submission contain copyrighted images. All PLOS content is published under the Creative Commons Attribution License (CC BY 4.0), which means that the manuscript, images, and Supporting Information files will be freely available online, and any third party is permitted to access, download, copy, distribute, and use these materials in any way, even commercially, with proper attribution. For more information, see our copyright guidelines: http://journals.plos.org/plosone/s/licenses-and-copyright.

 1. You may seek permission from the original copyright holder of Figure 9,13 and 17 to publish the content specifically under the CC BY 4.0 license.

Reviewers' comments:

Reviewer's Responses to Questions

**Comments to the Author**

1. Is the manuscript technically sound, and do the data support the conclusions?

Reviewer #1: Yes

Reviewer #2: Yes

Reviewer #3: Yes

2. Has the statistical analysis been performed appropriately and rigorously? 

Reviewer #1: N/A

Reviewer #2: N/A

Reviewer #3: Yes

3. Have the authors made all data underlying the findings in their manuscript fully available?

Reviewer #1: Yes

Reviewer #2: Yes

Reviewer #3: Yes

4. Is the manuscript presented in an intelligible fashion and written in standard English?

Reviewer #1: Yes

Reviewer #2: Yes

Reviewer #3: Yes

5. Review Comments to the Author

Reviewer #1: REVIEW

of the Research Article entitled: "Structural Optimization of the Excavator Boom Under Extreme Working Conditions Using EDEM–ADAMS Coupled Simulation: Structural optimization design" (PONE-D-25-31404)

Authors: Huawei Wu, Xiaolong Ding, Gui Liu, Xiaoyuan Zhu, Hualiang Wang

The reviewed article contains 34 pages, 5 chapters, 21 figures, 28 formulas, 9 tables and 17 literature sources.

Тhe title of this study is correctly selected and brings new elements to this topic, because: Hydraulic excavators are extremely widely used machines in the mining, construction, agricultural and infrastructure sectors. Therefore, it is natural to strive for their continuous improvement in terms of structure, topology, strength characteristics, power capabilities, reliability and durability, under different operating conditions and when performing different tasks.

In the Abstract, the authors consistently describe the stages of the research. The methods and means for achieving the set goals are shown. The results obtained and their significance are correctly described.

In the Introduction, the authors review most of the latest and most relevant research on the above issues. Seventeen sources published in the last 5-6 years have are reviewed. It is found that recent research focuses mainly on the analysis of excavation resistance and digging force, with the development of dynamic models, using simulations and topological optimizations. Recent advances in virtual prototyping facilitate the modeling of coupled dynamics and discrete elements for the analysis of excavator performance.

Based on the review, the authors set themselves the goal of filling the missing link in the research, namely to create a systematic methodology for generating load spectra under multiple conditions, for identifying critical operating conditions, for optimizing strength and stiffness for structural improvement.

In Chapter II a kinematic analysis and dynamic modeling of excavation mechanisms are performed. For the purposes of this study, a multifunctional small machine (an excavator for excavation works) is designed and manufactured and a method for optimization under extreme operating conditions is proposed. The excavator’s manipulator is implemented according to the classical scheme: boom, arm and booked, driven by hydraulic cylinders and operating exclusively in the vertical plane.

The kinematic model of the three-linkage mechanism is constructed using the standard Denavit–Hartenberg (D-H) method. The D-H parameters and the homogeneous transformation matrix from the base frame to the bucket-end frame are correctly described. Forward kinematic computations for the determination of the basic angles θ are arranged based on the cosine formulae. This way all dimensions and angles of the excavator’s manipulator are described.

The proposed model of virtual prototyping enables rapid acquisition of hinge joint load data, significantly enhancing efficiency while reducing experimental resource expenditure. The geometric model is imported into Adams for multibody dynamics simulation to establish linkage kinematic relationships.

After elimination of redundant degrees of freedom and overconstraints, 22 essential kinematic joints are applied at critical hinge locations.

Based on the operating parameters of each cylinder and the kinematic analysis of the working conditions a rigid-body dynamics model is constructed and validated, from which the working range diagram under typical no-load conditions is obtained to verify the forward kinematic solution.

In chapter III - Development and Operating Condition Analysis of an EDEM-Coupled Model, the enhanced discrete element method (EDEM) is coupled with the multi-body dynamics simulation software ADAMS to effectively analyze the digging operation and to monitor in real time the contact forces between the bucket and the material, as well as the load variations at the hinge joints of the connecting rods.

An EDEM model is built for the mechanism and ore particles, describing contact parameters between materials - such as the static friction coefficient, rolling friction coefficient, and coefficient of restitution. Five models of particle unit are used – ellipsoidal, triangular pyramid, cylindrical, cubic and triangular prism.

A coupled EDEM–Adams simulation is conducted to evaluate two typical operating conditions: front digging and lateral digging. The corresponding digging resistance curves of the bucket under both scenarios are presented in excavation resistance diagrams. The load diagrams (force/time) are constructed for the four main boom joints ( Joints 11, 12, 13 and 14) under both digging conditions.

The simulation results indicate that, during the 15-seconds operation cycle, the hinge point loads under forward excavation exhibit two distinct peaks: one at 7.65 s and another at 8.9 s, which correspond to the peak moments of the digging resistance. In contrast, the peak velocity under lateral excavation appears at 7.96 s, which aligns with its corresponding peak load phase. The three positions described above are considered extreme working conditions with the heaviest load on the boom joints.

Based on the hydraulic cylinder driving functions in ADAMS and the cylinder stroke ranges, the load of each joint of the boom under the extreme loading condition is determined, and shown on the diagrams.

After completing the simulation setup, the equivalent stress contours and the total deformation distribution of the boom under each operating condition are obtained and shown on the diagrams.

The results of the finite element analysis (FEA) reveals significant differences in the mechanical response of the boom under three sets of extreme loading conditions. The forward excavation condition at t = 8.9 s represents the most critical scenario, exhibiting a peak equivalent stress and a maximum deformation. The lateral excavation condition at t = 7.96 s yields the lowest mechanical response.

In chapter IV a design optimization of the boom and comparative validation is performed.

The study employs topology optimization to achieve improved performance and lightweight design. The structural design task is as a material distribution problem, to maximize stiffness and minimize deformation. An objective function design variable and constraint conditions are used.

Images of boom topology optimization under extreme working conditions are shown before and after optimization.

To verify the reliability of the topology optimized boom model, a static simulation analysis is conducted on the boom under the same excavation conditions, applying identical loading and boundary conditions, with the results shown in stress and deformation diagrams.

The results after the boom optimization are very good: the mass is reduced by 13.8%, the stress by 26.12%, and the deformation by 29.11%.

Finally, a validation and comparison of the optimized design is made. To evaluate the effectiveness of the boom optimization under the most critical working condition identified through EDEM–ADAMS co-simulation, the optimized design is compared with the results obtained using a conventional optimization approach. The validation showed that the proposed method gives better results than the conventional optimization approach.

In the Conclusion, the authors show the contributions of this work and the applied method, defending them with the results obtained.

About the present article I can mention the following technical errors and remarks:

- in D-H transformation matrix (equation 1), in the first row, third column (i.e. element 1,3) there must be “sin.sin” instead of “sin.cos”;

- on page 33, after Fig.16, on the fourth row is written “Fig.3.15” – this figure does not exist;

- on page 20 is written “scrap-and-dig15” – what does 15 mean?;

- In the Fig.15. – e) and f), there must be 7,96 s.

- In the Table 3 the kinematic joints can be described more precisely;

- After the formulae (28) - the objective function f(x) can be described in more details with corresponding criteria.

CONCLUSION

The reviewed paper is obviously an original research paper with important topic, technical rigorous and meets the scientific and ethical standard for inclusion in the published scientific report.

This paper proposes a unique method for structural optimization of the extreme loading conditions of the excavator mechanism in dynamic environment.

High level of analysis and optimization methods are applied such as kinematic analysis of the D-H coordinates, the ADAMS multi-body dynamics software, EDEM–ADAMS Coupled Simulation. After determining the extreme operating conditions of the boom, force and deformation optimization is performed for these conditions, in order to achieve greater strength and lower deformations at a reduced weight of the excavator.

The conclusion is correctly done supported by the research results.

In conclusion, I recommend that this work be included for publication after the aforementioned technical errors are corrected.

Reviewer #2: The manuscript is well-structured and follows a logical progression from problem definition to methodology, results, and conclusions. The integration of EDEM–ADAMS coupling with topology optimization is a solid contribution, especially for excavator boom optimization under extreme conditions.

However, there are areas where the work could be clearer, more rigorous in validation, and more concise.

While the method is well-integrated, similar approaches (DEM–MBD + FEA optimization) exist in literature for other components. The novelty is more in application than in fundamental methodology.

1. Certain figures (e.g., FEA stress distributions) do not include numerical scale bars or units, which reduces clarity and makes interpretation more difficult.

2. Terminology Consistency – The manuscript uses multiple terms such as “pick-and-dredge,” “digging mechanism,” and “excavating mechanism” interchangeably without clear definitions, which may confuse readers.

3. Kinematics Section Detail – The derivations in the kinematics section are presented in excessive detail for the main text and could be shortened or moved to an appendix to improve readability.

4. No experimental validation – While simulations are comprehensive, there is no physical testing to confirm load cases, deformation magnitudes, or optimization effects.

5. The manuscript requires minor English revision in addition to the correction of typographical errors, such as "Loud image" instead of "Load image" in Fig. 18 caption, emoving the stray number in ‘scrap-and-dig15 transporter, etc.

Reviewer #3: Comments for improvement:

Section II-A: Prototype Development: Remove the word “diagram” from Fig. 1. Figure caption should be like “Fig. 1 Structural Assembly of the Loader-Excavator”.

Section II-A: Prototype Development: Please keep figure for the top view of the Loader-Excavator and change the figure numbers accordingly.

Section II-C: Forward kinematics formulation: In the present study the working range diagram under typical no-load conditions is obtained to verify the forward kinematic solution. But the working range represents the working space through which the bucket tip can reach within in this range. The simulation is also one of the tool to validate the forward kinematics but need to show case the comparison of the input joint angles with bucket tip pose. Recommended to perform comparison of results of simulation with forward kinematics or perform inverse kinematics to validate the forward kinematic model developed for loader-excavator.

In the section II-D, it is mentioned that “Following elimination of redundant degrees of freedom and overconstraints, essential kinematic joints were applied at critical hinge locations” but its methodology is missing to achieve the same. Please add the methodology or justify clearly that how it was achieved.

In Fig. 6: If possible increase the font size of the nomenclatures given to all the joints for better readability.

Fig. 7: Scope of work diagram: Generally, best suitable name for this figure 7 is “Working range diagram” in place of “scope of work diagram”.

Fig. 7: Presented working range diagram is overlapping with the shovel front of the loader-excavator. It is also visible in the Fig. 1 and Fig. 7. Might be the presented working range not be achieved as per the visual presentation given in the manuscript. Please justify exactly.

Table-6: Total mass of 780 kg and 1200 kg taken for particle types. Please justify that why different values taken for study?

Table-7: row -1, write F_z (N) in place of Fz (N). Do correction.

Table-8: row-1, unit of Modulus of elasticity should be MPa in place of Mpa. Do correction.

Para-1 (Content after the Fig. 16): Presently written last line: “This kinematic configuration corresponds spatially to the maximum digging depth condition illustrated in Fig. 3.15”.

Here, wrong figure number is written. Please write correct figure number. If might be Fig. 16.

Paragraph above Fig. 16 and paragraph below Fig. 17: To avoid repetition of statements.

[See Paragraph above Fig. 16: Notably, the forward excavation condition at t = 8.9 s represents the most critical scenario, exhibiting a peak equivalent stress of 131.42 MPa and a maximum deformation of 0.10777 mm. In contrast, the lateral excavation condition at t = 7.96 s yields the lowest mechanical response, with a peak equivalent stress of 104.13 MPa and a deformation of 0.072722 mm.]

[See Paragraph below Fig. 17: Finite element analysis of the boom under three sets of extreme loading conditions reveals that the equivalent stress and total deformation reach their maximum values at 8.9 s under the forward excavation condition, with respective peaks of 131.42 MPa and 0.10777 mm.

In contrast, the minimum mechanical response occurs at 7.96 s under the lateral excavation

condition, where the equivalent stress and total deformation are 104.13 MPa and 0.07272 mm,

respectively.]

Section IV: Design Optimization of the Boom and Comparative Validation: 3rd paragraph, “The boundary condition is defined as a mass retention ratio of 70%”. On what basis or consideration, the mass retention ratio taken as 70%? Justify.

Section IV-B: Validation and Comparison of the Optimized Design: In the present case, the conventional optimization approach adopted for comparison of equivalent stress and total deformation of optimized model of boom. There are two conventional approaches for optimization: size optimization and shape optimization. But in the present study, it is not defined that which approach adopted for conventional optimization (i.e. size or shape optimization) for comparison purposes. Also study does not shows any variation in thickness of plates or shapes of the boom. According to the given information in the manuscript, it reflects that only simple FEA performed for the boom and results were compared for equivalent stress and total deformation.

Please elaborate the justification perfectly and revise the content accordingly.

In the conclusion section: Add the future scope of the presented study. (What kind of extension of the study presented is possible?).

This will provide motivation and scope to the readers of the journal for their future research work.

Note: Minor Revision is Required. It is required to incorporate/justify the given comments.

6. PLOS authors have the option to publish the peer review history of their article (what does this mean? ). If published, this will include your full peer review and any attached files.

**Do you want your identity to be public for this peer review?** For information about this choice, including consent withdrawal, please see our Privacy Policy .

Reviewer #1: **Yes: ** Krasimir Ganchev

Reviewer #2: No

Reviewer #3: **Yes: ** Dr. Bhaveshkumar P. Patel

---

## [Author Response · Author response to Decision Letter 1]

11 Oct 2025

1.Point-by-point response to Comments and Suggestions for Editor

Comments 1: Please ensure that your manuscript meets PLOS ONE's style requirements, including those for file naming.

Response 1: Thank you for pointing this out. I have revised the article's format to meet the journal's requirements. Details are provided in the newly submitted manuscript.

Comments 2: We note that the grant information you provided in the ‘Funding Information’ and ‘Financial Disclosure’ sections do not match.

Response 2: Thank you for pointing this out. I have updated the corresponding funding fund and number. Detailed information is provided in the newly submitted manuscript.

Comments 3: Please amend either the title on the online submission form (via Edit Submission) or the title in the manuscript so that they are identical.

Response 3: Thank you for pointing this out.I have changed the title of the online form to match the manuscript title.

Comments 4: Please amend the manuscript submission data (via Edit Submission) to include author Gui Liu, Xiaoyuan Zhu and Hualiang Wang.

Response 4: Thank you for pointing this out. I have completed the author contributions and corresponding institutional information. Detailed information is shown in the newly submitted manuscript.

Comments 5: We note that Figure 9,13 and 17 in your submission contain copyrighted images. All PLOS content is published under the Creative Commons Attribution License (CC BY 4.0), which means that the manuscript, images, and Supporting Information files will be freely available online, and any third party is permitted to access, download, copy, distribute, and use these materials in any way, even commercially, with proper attribution.

 If you are unable to obtain permission from the original copyright holder to publish these figures under the CC BY 4.0 license or if the copyright holder’s requirements are incompatible with the CC BY 4.0 license, please either i) remove the figure or ii) supply a replacement figure that complies with the CC BY 4.0 license. Please check copyright information on all replacement figures and update the figure caption with source information. If applicable, please specify in the figure caption text when a figure is similar but not identical to the original image and is therefore for illustrative purposes only.

Response 5: Thank you for pointing this out. I have provided a replacement photo that is compliant with the license. Details are shown in the new submitted manuscript.

2. Point-by-point response to Comments and Suggestions for Reviewer#1

Comments 1: in D-H transformation matrix (equation 1), in the first row, third column (i.e. element 1,3) there must be “sin.sin” instead of “sin.cos”.

Response 1: Thank you for pointing this out. I have corrected this technical error by changing the "sin.cos" in Equation (1) to "sin.sin". It appears in Equation (1) on line 112 of the revised manuscript with track changes.

Comments 2: on page 33, after Fig.16, on the fourth row is written “Fig.3.15” – this figure does not exist.

Response 2: Thank you for pointing this out. I have corrected this technical error by adding the relevant kinematic analysis verification and changing the correct figure number to "Fig.18". The details are shown in lines 375-381 and 544 of the revised manuscript with track changes.

Comments 3: on page 20 is written “scrap-and-dig15” – what does 15 mean?

Response 3: Thank you for pointing this out. Due to a typographical error that caused this technical issue, I have corrected it by removing the number 15 and standardizing the professional terminology used in this study. Detailed information is provided on line 395 of the revised manuscript with track changes.

Comments 4: In the Fig.15. – e) and f), there must be 7,96 s.

Response 4: Thank you for pointing this out. I have made the corresponding legend modification, which is shown in lines 485 and 528-530 of the revised manuscript with track changes.

Comments 5: In the Table 3 the kinematic joints can be described more precisely.

Response 5: Thank you for pointing this out. I have made detailed revisions to the hinge motion pairs in Table 3. To accommodate additional requirements for dynamic simulation verification, an extra table has been added. The original Table 3 has been updated to become Table 4, which now provides precise descriptions of all hinge motion pairs, including their numbers, names, and motion pair types. Detailed information is available in line 300 of the revised manuscript with track changes under Table 4.

Comments 6: After the formulae (28) - the objective function f(x) can be described in more details with corresponding criteria.

Response 6: Thank you for your valuable suggestions to improve the paper. I have refined the objective function formula with detailed explanations for each parameter. These enhancements are clearly shown in lines 571-580 of the revised manuscript with track changes. Your expert feedback has greatly enhanced the completeness of my work. I sincerely appreciate all the constructive advice you provided.

3. Point-by-point response to Comments and Suggestions for Reviewer #2

Comments 7: Certain figures (e.g., FEA stress distributions) do not include numerical scale bars or units, which reduces clarity and makes interpretation more difficult.

Response 7: Thank you for pointing this out. I have reviewed all images and corrected those that should have been labeled with units or scales but weren't. Detailed information is provided in lines 594-596 of the revised manuscript with track changes. Thank you very much for your valuable feedback, which helped identify technical errors in the article.

Comments 8: Terminology Consistency – The manuscript uses multiple terms such as “pick-and-dredge,” “digging mechanism,” and “excavating mechanism” interchangeably without clear definitions, which may confuse readers.

Response 8: Thank you very much for your valuable feedback. I have standardized the technical terminology in the article by uniformly correcting it to "backhoe loader". The detailed modifications are marked in the revised manuscript with track changes at lines 26,29,41,42,99,109,123,133,138,212,221,301,395,406,441,445,560, and 660.

Comments 9: Kinematics Section Detail – The derivations in the kinematics section are presented in excessive detail for the main text and could be shortened or moved to an appendix to improve readability.

Response 9: Thank you for pointing this out. I have streamlined the unnecessary derivations in the kinematic analysis while retaining only the essential ones. Detailed information is provided in the revised manuscript with track changes at lines 148-157,164-179, and 186-210.

Comments 10: No experimental validation – While simulations are comprehensive, there is no physical testing to confirm load cases, deformation magnitudes, or optimization effects.

Response 10: Thank you very much for your valuable suggestions. While your recommendations demonstrate strong guidance and professional expertise, we currently lack sufficient funding and laboratory conditions to conduct physical experiments for data validation. Our literature review indicates that simulation results closely match experimental data, effectively replicating object motion processes. Detailed information is available in the revised manuscript with track changes (lines 391-393). We will enhance communication with our supervisor to secure additional funding for focused research on novel paper development and related physical experiments. We sincerely appreciate your continued support, as your feedback will guide our ongoing efforts to improve research quality.

Comments 11: The manuscript requires minor English revision in addition to the correction of typographical errors, such as "Loud image" instead of "Load image" in Fig. 18 caption, emoving the stray number in ‘scrap-and-dig15 transporter, etc.

Response 11: Thank you for pointing this out. I have corrected the English errors in the article. They appear on lines 278,395,444 and 591 of the revised manuscript with track changes.

4. Point-by-point response to Comments and Suggestions for Reviewer #3

Comments 12: Section II-A: Prototype Development: Remove the word “diagram” from Fig. 1. Figure caption should be like “Fig. 1 Structural Assembly of the Loader-Excavator”.

Response 12: Thank you for pointing this out. I have changed the name of Fig 1. It appears on line 99 in the revised manuscript with track changes.

Comments 13: Section II-A: Prototype Development: Please keep figure for the top view of the Loader-Excavator and change the figure numbers accordingly.

Response 13: Thank you for pointing this out. I have added the top view and front view with corresponding fig numbers. It is shown in lines 93-99 of the revised manuscript with track changes.

Comments 14: Section II-C: Forward kinematics formulation: In the present study the working range diagram under typical no-load conditions is obtained to verify the forward kinematic solution. But the working range represents the working space through which the bucket tip can reach within in this range. The simulation is also one of the tool to validate the forward kinematics but need to show case the comparison of the input joint angles with bucket tip pose. Recommended to perform comparison of results of simulation with forward kinematics or perform inverse kinematics to validate the forward kinematic model developed for loader-excavator.

Response 14: Thank you very much for your valuable suggestions. Your feedback will be crucial in optimizing my article. I have added comparative analysis between input joint angles and bucket tip positions to validate the comparison between kinematic analysis and simulation results. Additionally, I corrected errors in cylinder stroke calculations and angle computations from the kinematic analysis. Finally, by comparing data obtained from both methods, it was shown that the discrepancies between kinematic analysis and simulation analysis were minimal, confirming the reliability of both approaches. Detailed information is available in lines 217-277 and 308-387 of the revised manuscript with track changes. Once again, I appreciate your suggestions as they have significantly enhanced the

Comments 15: In the section II-D, it is mentioned that “Following elimination of redundant degrees of freedom and overconstraints, essential kinematic joints were applied at critical hinge locations” but its methodology is missing to achieve the same. Please add the methodology or justify clearly that how it was achieved.

Response 15: Thank you for pointing this out. The method for eliminating redundant degrees of freedom has been supplemented. Details are shown in lines 285-295 of the revised manuscript with track changes.

Comments 16: In Fig. 6: If possible increase the font size of the nomenclatures given to all the joints for better readability.

Response 16: Thank you for pointing this out. I have made the image modifications shown in Fig 9, lines 298-299 of the revised manuscript with track changes.

Comments 17: Fig. 7: Scope of work diagram: Generally, best suitable name for this figure 7 is “Working range diagram” in place of “scope of work diagram”.

Response 17: Thank you for pointing this out. I have revised the legend, which is shown in line 310 of the revised manuscript with track changes.

Comments 18: Fig. 7: Presented working range diagram is overlapping with the shovel front of the loader-excavator. It is also visible in the Fig. 1 and Fig. 7. Might be the presented working range not be achieved as per the visual presentation given in the manuscript. Please justify exactly.

Response 18: Thank you for pointing this out. The presence of material baffles may introduce visual inaccuracies in the working range diagram. However, during actual operation, the mining device did not exhibit motion interference with other components. I have revised the diagram by adding detailed zoom-in views to demonstrate simulation results from multiple angles. As shown in Fig 1, no motion interference occurred. Detailed information is provided in lines 309-310 (Fig 10) and lines 92-99 (Fig 1) of the revised manuscript with track changes.

Comments 19: Table-6: Total mass of 780 kg and 1200 kg taken for particle types. Please justify that why different values taken for study?

Response 19: Thank you for pointing this out. Based on existing literature, I have supplemented the rationale for particle type configuration and corrected errors in the original article's particle mass measurement results. Detailed information is provided in lines 413-418 of the revised manuscript with track changes and in Table 8 on line 426.

Comments 20: Table-7: row -1, write F_z (N) in place of Fz (N). Do correction.

Response 20: Thank you for pointing this out. I have corrected the corresponding technical error, which appears in line 492 of table 9, row 1, of the revised manuscript with track changes.

Comments 21: Table-8: row-1, unit of Modulus of elasticity should be MPa in place of Mpa. Do correction.

Response 21: Thank you for pointing this out. I have corrected the corresponding English error, which appears in line 499 of table 10, row 1, of the revised manuscript with track changes.

Comments 23: Para-1 (Content after the Fig. 16): Presently written last line: “This kinematic configuration corresponds spatially to the maximum digging depth condition illustrated in Fig. 3.15”.

Here, wrong figure number is written. Please write correct figure number. If might be Fig. 16.

Response 23: Thank you for pointing this out. I have corrected this technical error by adding the relevant kinematic analysis verification and changing the correct figure number to "Fig.18". The details are shown in lines 375-381 and 543 of the revised manuscript with track changes.

Comments 24: Paragraph above Fig. 16 and paragraph below Fig. 17: To avoid repetition of statements.

[See Paragraph above Fig. 16: Notably, the forward excavation condition at t = 8.9 s represents the most critical scenario, exhibiting a peak equivalent stress of 131.42 MPa and a maximum deformation of 0.10777 mm. In contrast, the lateral excavation condition at t = 7.96 s yields the lowest mechanical response, with a peak equivalent stress of 104.13 MPa and a deformation of 0.072722 mm.]

[See Paragraph below Fig. 17: Finite element analysis of the boom under three sets of extreme loading conditions reveals that the equivalent stress and total deformation reach their maximum values at 8.9 s under the forward excavation condition, with respective peaks of 131.42 MPa and 0.10777 mm.

In contrast, the minimum mechanical response occurs at 7.96 s under the lateral excavation

condition, where the equivalent stress and total deformation are 104.13 MPa and 0.07272 mm,

respectively.]

Response 24: Thank you for pointing this out. I have removed the duplicate statement. Details are shown in lines 551-558 of the revised manuscript with track changes.

Comments 25: Section IV: Design Optimization of the Boom and Comparative Validation: 3rd paragraph, “The boundary condition is defined as a mass retention ratio of 70%”. On what basis or consideration, the mass retention ratio taken as 70%? Justify.

Response 25: Thank you for pointing this out. I have added a basis and consideration for a 70% mass retention rate based on the suggestion and existing literature. Detailed information is shown in lines 583-585 of the revised manuscript with track changes.

Comments 26: Section IV-B: Validation and Comparison of the Optimized Design: In the present case, the conventional optimization approach adopted for comparison of equivalent stress and total deformation of optimized model of boom. There are two conventional approaches for optimization: size optimization and shape optimization. But in the present study, it is not defined that which approach adopted for conventional optimization (i.e. size or shape optimization) for comparison purposes. Also study does not shows any variation in thickness of plates or shapes of th

---

## [Decision Letter · Decision Letter 1]

10 Nov 2025

Structural optimization of the excavator boom under extreme working conditions using EDEM–ADAMS coupled simulation

PONE-D-25-31404R1

Dear Dr. 丁,

We’re pleased to inform you that your manuscript has been judged scientifically suitable for publication and will be formally accepted for publication once it meets all outstanding technical requirements.

Kind regards,

Carlos Alberto Cruz-Villar, Ph. D.

Academic Editor

PLOS ONE

Additional Editor Comments: The Academic Editor grants acceptance under the understanding that the Authors will comply with the PLOS One criteria for papers describing new methods or software for applications. Specifically, these reports must meet the criteria of utility, validation, and availability, which are described in detail at http://journals.plos.org/plosone/s/submission-guidelines#loc-methods-software-databases-and-tools.

Reviewers' comments:

Reviewer's Responses to Questions

**Comments to the Author**

1. If the authors have adequately addressed your comments raised in a previous round of review and you feel that this manuscript is now acceptable for publication, you may indicate that here to bypass the “Comments to the Author” section, enter your conflict of interest statement in the “Confidential to Editor” section, and submit your "Accept" recommendation.

Reviewer #1: All comments have been addressed

Reviewer #3: All comments have been addressed

2. Is the manuscript technically sound, and do the data support the conclusions?

Reviewer #1: Yes

Reviewer #3: Yes

3. Has the statistical analysis been performed appropriately and rigorously? 

Reviewer #1: N/A

Reviewer #3: Yes

4. Have the authors made all data underlying the findings in their manuscript fully available?

Reviewer #1: Yes

Reviewer #3: Yes

5. Is the manuscript presented in an intelligible fashion and written in standard English?

Reviewer #1: Yes

Reviewer #3: Yes

6. Review Comments to the Author

Reviewer #1: The authors have satisfactorily addressed all my comments. I recommend this manuscript for publication.

Reviewer #3: A kinematic modeling and structural optimization analysis were performed for a Loader-Excavator as a digging transportation device, and attempt is made to improve the performance of boom under critical working condition. The dynamic simulation and working condition identified using the ADAMS software. As well as excavation resistance and hinge-point loads were obtained using EDEM-based excavation simulation. Ultimately achieved the reduction in weight of the boom, deformation and equivalent stresses through structural optimization. Also further reduced equivalent stress and deformation in comparison with the model optimized by conventional method.

I appreciate the efforts made by authors for the current state of the manuscript presented. The abstract is perfectly written. The flow of content presented in the manuscript is perfect and technically correct. Right methodology adopted to justify the title of the manuscript and objectives.

Within the revised manuscript, authors have incorporated all suggested comments and now no further correction is required in the current state of manuscript.

It is recommended to accept the manuscript for publication.

7. PLOS authors have the option to publish the peer review history of their article (what does this mean? ). If published, this will include your full peer review and any attached files.

**Do you want your identity to be public for this peer review?** For information about this choice, including consent withdrawal, please see our Privacy Policy .

Reviewer #1: No

Reviewer #3: No

---

## [Editor Report · Acceptance letter]

PONE-D-25-31404R1

PLOS ONE

Dear Dr. Ding,

I'm pleased to inform you that your manuscript has been deemed suitable for publication in PLOS ONE. Congratulations! Your manuscript is now being handed over to our production team.

Kind regards,

on behalf of

Dr. Carlos Alberto Cruz-Villar

Academic Editor

PLOS ONE